# The Vertex-Attribute-Constrained Densest $k$-Subgraph Problem

**Qiheng Lu**                                                                *luqh@virginia.edu*
*University of Virginia*

**Nicholas D. Sidiropoulos**                                                  *nikos@virginia.edu*
*University of Virginia*

**Aritra Konar**                                                         *aritra.konar@kuleuven.be*
*KU Leuven*

Reviewed on OpenReview: *https://openreview.net/forum?id=ae8hda3atq*

## Abstract

Dense subgraph mining is a fundamental technique in graph mining, commonly applied in fraud detection, community detection, product recommendation, and document summarization. In such applications, we are often interested in identifying communities, recommendations, or summaries that reflect different constituencies, styles or genres, and points of view. For this task, we introduce a new variant of the Densest $k$-Subgraph (D$k$S) problem that incorporates the attribute values of vertices. The proposed *Vertex-Attribute-Constrained Densest k-Subgraph* (VAC-D$k$S) problem retains the NP-hardness and inapproximability properties of the classical D$k$S. Nevertheless, we prove that a suitable continuous relaxation of VAC-D$k$S is tight and can be efficiently tackled using a projection-free Frank–Wolfe algorithm. We also present an insightful analysis of the optimization landscape of the relaxed problem. Extensive experimental results demonstrate the effectiveness of our proposed formulation and algorithm, and its ability to scale up to large graphs. We further elucidate the properties of VAC-D$k$S versus classical D$k$S in a political network mining application, where VAC-D$k$S identifies a balanced and more meaningful set of politicians representing different ideological camps, in contrast to the classical D$k$S solution which is unbalanced and rather mundane.

## 1 Introduction

Dense subgraph detection is a fundamental graph mining primitive that aims to identify highly connected subsets of vertices in a given graph. It has found widespread applications, including fraud detection in consumer reviews, product recommendation, and financial transaction networks (Hooi et al., 2016; Li et al., 2020; Ji et al., 2022; Chen & Tsourakakis, 2022), as well as community detection in social networks, topic mining (Angel et al., 2014), and gene association studies (Saha et al., 2010).

Many applications, such as team formation and polarization detection, can benefit from incorporating vertex attribute values into the formulation (Adami et al., 2011; Gajewar & Das Sarma, 2012; Fazzone et al., 2022). Recently, several studies (Gajewar & Das Sarma, 2012; Anagnostopoulos et al., 2020; 2024; Miyauchi et al., 2023; Kariotakis et al., 2025) have proposed various vertex-attribute-aware dense subgraph mining formulations and algorithms. In this work, we address limitations in the problem formulations and algorithms of existing studies on attribute-constrained dense subgraph mining, which are discussed in detail in Section 1.1.

There exist multiple pertinent formulations of dense subgraph mining (see the survey (Lanciano et al., 2024) and references therein). One of the more prominent is the *Densest Subgraph* (DSG) problem (Goldberg, 1984) which aims to extract the subgraph with the maximum average induced degree. DSG can be solved

exactly via maximum-flow, and linear time greedy algorithms backed by approximation guarantees are also available (Charikar, 2000; Boob et al., 2020; Chekuri et al., 2022). In addition, recent work (Danisch et al., 2017; Harb et al., 2022; Nguyen & Ene, 2024) has developed a suite of convex optimization algorithms for solving the problem. An alternative is to seek a subgraph that maximizes the minimum (instead of the average) induced degree, which is known as the $k$-core of a graph (Seidman, 1983).

A drawback of DSG and $k$-core is that they often yield large but loosely connected subgraphs (Tsourakakis et al., 2013; Shin et al., 2016). A remedy that affords explicit control of subgraph size is the *Densest $k$-Subgraph* (D$k$S) problem (Feige et al., 2001), which seeks a subset of $k$ vertices with the maximum number of edges between them. However, D$k$S is NP-hard and difficult to approximate in the worst case (Khot, 2006; Manurangsi, 2017). The best polynomial-time approximation algorithm for D$k$S offers an $O(n^{1/4+\epsilon})$-approximation at complexity $O(n^{1/\epsilon})$ for $\epsilon > 0$ (Bhaskara et al., 2010). In light of the problem's difficulty, different convex relaxations of D$k$S have been considered. For example, the work of (Ames, 2015; Bombina & Ames, 2020) considered relaxations based on Semidefinite Programming (SDP), but these are a heavy lift in terms of computation. Various "lightweight" continuous relaxations of D$k$S have also been pursued, including the gradient-based approaches (Hager et al., 2016; Sotirov, 2020; Liu et al., 2024) and the more involved Lovász-ADMM approach (Konar & Sidiropoulos, 2021). However, the tightness of these relaxations has not been investigated. Recently, Lu et al. (2025b) proposed a provably tight continuous relaxation formulation. In the extended version of Lu et al. (2025b), Lu et al. (2025a) analyzed the optimization landscape to demonstrate the advantages of this formulation. Furthermore, the Frank–Wolfe-based algorithm proposed by Lu et al. (2025a;b) has shown strong performance in both solution quality and scalability.

A very different approximation approach to D$k$S relative to all the above, promoted by Papailiopoulos et al. (2014), is to use a low-rank surrogate of the graph adjacency matrix to leverage the so-called *Spannogram*—a low-rank "geometric" solver for certain bilinear quadratic optimization problems. In practice, using rank as low as two entails complexity $O(n^3)$, which is a challenge for large-scale problems. This approach is therefore essentially limited to using a rank-one approximation. The approach of Papailiopoulos et al. (2014) also provides for a simple upper bound on the optimal edge density of D$k$S, which gives us a problem-instance-dependent approximation gap bound.

## 1.1 Related Work: Attribute-Constrained Dense Subgraph Mining

While dense subgraph discovery is a well-studied topic, only recently has the problem of extracting vertex-attribute-constrained dense subgraphs gained attention (Gajewar & Das Sarma, 2012; Anagnostopoulos et al., 2020; 2024; Miyauchi et al., 2023; Kariotakis et al., 2025). These works are motivated by the fact that subgraphs extracted via DSG or its variants may violate attribute-based requirements, as such formulations do not explicitly consider vertex attributes. To address this limitation, Gajewar & Das Sarma (2012); Anagnostopoulos et al. (2020; 2024); Miyauchi et al. (2023); Kariotakis et al. (2025) have proposed formulations and algorithms that incorporate vertex-attribute constraints into dense subgraph mining. These were among the first efforts to introduce vertex-attribute constraints into the task. Nonetheless, the area remains largely underexplored, with many key challenges still open.

Anagnostopoulos et al. (2020; 2024) proposed a vertex-attribute-constrained variant of DSG, where each vertex belongs to a group, and the objective is to identify a subgraph with maximum average induced degree that includes an equal number of vertices from each group. The spectral relaxation algorithms introduced in these works offer meaningful approximation guarantees, provided the degree distribution of the input graph is approximately uniform. However, real-world graphs often exhibit highly skewed degree distributions (Newman, 2003), and the theoretical guarantees apply only when the vertex attribute takes on exactly two distinct values.

A subsequent formulation, introduced as Problem 1 in Miyauchi et al. (2023), extends the setting of Anagnostopoulos et al. (2020; 2024) by allowing a variable minimum representation level across groups within the selected subgraph (i.e., a lower bound on the proportion of selected vertices belonging to each group). While this formulation represents a meaningful generalization, it still has two notable limitations. First, it enforces a uniform minimum representation level across all groups, which restricts the flexibility to specify different representation requirements based on application needs. Second, as it is based on the DSG framework, the

extracted subgraphs tend to be large but loosely connected—a known drawback of DSG-based formulations (Tsourakakis et al., 2013).

To solve the problem, Miyauchi et al. (2023) proposed a two-stage $\Omega(1/\sqrt{n})$-approximation algorithm for this formulation. In the first stage, an initial solution is obtained using a Densest-at-least-$k$-Subgraph (Dal$k$S) algorithm with a known approximation guarantee—either $1/3$ or $1/2$, depending on the specific method (Andersen & Chellapilla, 2009; Khuller & Saha, 2009). This solution is then refined in the second stage through a post-processing procedure that incrementally adds vertices until the attribute constraint is satisfied. Since the attribute constraints enforced in the post-processing stage restrict the feasible solution space, the optimal edge density under these constraints is already no greater than that of the Dal$k$S problem considered in the first stage. The fact that the approximation ratio further drops—from a constant factor to $\Omega(1/\sqrt{n})$—suggests that the post-processing step may significantly compromise the edge density of the resulting subgraph.

An alternative DSG-based formulation models attribute constraints by explicitly specifying a minimum number of selected vertices from each group. This formulation was originally introduced by Gajewar & Das Sarma (2012) in the context of multi-skill collaborative team formation , and was more recently revisited by Miyauchi et al. (2023) as Problem 2 in their work. While this formulation offers greater flexibility, the proposed approximation algorithm suffers from poor scalability—reportedly requiring over 10,000 seconds to process a graph with only 126 vertices. In addition, this formulation also exhibits the same limitation as other DSG-based formulations, namely the tendency to produce large but loosely connected subgraphs. Because the size of the subgraph cannot be controlled, it is also not possible to ensure that the proportion of vertices selected from each group exceeds a non-trivial threshold.

To circumvent the NP-hardness of the formulations in prior work (Gajewar & Das Sarma, 2012; Anagnostopoulos et al., 2020; 2024; Miyauchi et al., 2023), Kariotakis et al. (2025) proposed two regularization-based formulations for incorporating vertex attributes which are solvable in polynomial time. However, like other DSG-based methods, their approach lacks explicit control over the size of the extracted subgraphs. Moreover, the approach only applies to the case of binary vertex attributes and requires bisection search to determine the regularization parameter that ensures that the extracted subgraph satisfies a desired representation level of the vertices.

We also note that a recent paper (Oettershagen et al., 2024) considered a variant of the DSG problem for networks with multiple *edge* (as opposed to vertex) attributes that represent different kinds of relationships between vertices. Oettershagen et al. (2024) proposed formulations for finding the densest subgraph that contains exactly, at most, or at least a specified number of edges for each edge attribute. They showed that the decision versions of these problems are NP-complete and developed a linear-time constant-factor approximation algorithm, which, however, only applies to *everywhere sparse* graphs—a restrictive assumption in the context of dense subgraph mining. To summarize, their formulation differs significantly from ours: it focuses on edge-attribute rather than vertex-attribute constraints and is based on DSG instead of D$k$S. Additionally, our theoretical analysis does not rely on the everywhere sparse assumption.

## 1.2 Our Contributions

The main contributions of this paper are fourfold:

- We propose a new variant of the Densest $k$-Subgraph problem, termed the Vertex-Attribute-Constrained Densest $k$-Subgraph (VAC-D$k$S) problem, which explicitly incorporates vertex attribute information into the subgraph selection process. Compared to existing approaches, our formulation enables explicit control over the subgraph size, as well as lower bounds on the number of selected vertices from each group. This prevents the extraction of large but loosely connected subgraphs and enables independent control over each group's selection, guaranteeing that its representation exceeds a non-trivial proportion.

- Although the VAC-D$k$S problem is NP-hard, we prove that a natural relaxation is tight and analyze the optimization landscape of the relaxed problem. Both results build upon, but constitute non-trivial generalizations of an analogous relaxation of the classical unweighted D$k$S problem studied

in Lu et al. (2025a). The main challenge is that the constraints of the relaxation of VAC-D$k$S are more involved, owing to the need to ensure that the representation level of each vertex group meets its target. Our key technical contribution is a more sophisticated two-stage rounding technique that is used to characterize the local and global maximizers of the relaxed problem in order to establish tightness and analyze the optimization landscape.

- To ensure scalability to large datasets, we seek efficient gradient-based methods to find high-quality solutions of the VAC-D$k$S relaxation. However, a key computational bottleneck is the cost of computing projections onto the constraint set during each iteration, which requires using general-purpose convex optimization solvers owing to the complex structure of the constraint set. To circumvent this bottleneck, we demonstrate that the projection-free Frank–Wolfe algorithm (Frank & Wolfe, 1956; Jaggi, 2013; Lacoste-Julien, 2016) is well-suited for the problem. It enables the computation of feasible ascent directions in *closed form*, which significantly reduces the computational cost of each iteration. We showcase its effectiveness in obtaining high-quality solutions and scalability across various scenarios.

- We demonstrate that our algorithm effectively uncovers more meaningful subgraphs with balanced political representation while simultaneously picking tone-setting politicians on a real-world Greek political network. Such an outcome is not attained by the classical D$k$S formulation, which tends to select ideologically skewed, less meaningful subsets that miss much of the political action.

## 1.3 Notation

In this paper, lowercase roman type letters, lowercase bold type letters, uppercase bold type letters, and uppercase calligraphic type letters denote scalars, vectors, matrices, and sets or pairs of sets, respectively. $[n]$ denotes the set $\{1, 2, \ldots, n\}$. $|\cdot|$ denotes the cardinality of a set. $x_i$ denotes the $i$-th entry of the vector $\boldsymbol{x}$. $a_{ij}$ denotes the entry in the $i$-th row and $j$-th column of matrix $\boldsymbol{A}$. $\boldsymbol{x}^{(t)}$ denotes the vector $\boldsymbol{x}$ at the $t$-th iteration. $\boldsymbol{x}^\top$ denotes the transpose of $\boldsymbol{x}$. $\text{top}_k(\boldsymbol{x}, \mathcal{C})$ denotes the index set of the top $k$ entries corresponding to the set index $\mathcal{C}$ in $\boldsymbol{x}$. $\boldsymbol{x}[\mathcal{C}] \leftarrow i$ denotes assigning the value $i$ to the entries corresponding to the index set $\mathcal{C}$ in $\boldsymbol{x}$.

## 2 Problem Statement

Consider a weighted, undirected, and simple graph $\mathcal{G} = (\mathcal{V}, \mathcal{E}, w)$ with at least one positive weight[1], where $\mathcal{V}$ is the set of $n = |\mathcal{V}|$ vertices and $\mathcal{E}$ is the set of $m = |\mathcal{E}|$ edges with weights defined by $w$. Let $\mathcal{C} = \{c_1, c_2, \ldots, c_r\}$ be a set of $r$ different attribute values and $\ell : \mathcal{V} \to \mathcal{C}$ be a mapping from a vertex to the corresponding attribute value. Each vertex in the graph $\mathcal{G}$ is assigned an attribute value from the set $\mathcal{C}$ by the mapping $\ell$. Let $\mathcal{C}_i = \{j \in \mathcal{V} \mid \ell(j) = c_i\}$ denote the set of vertices whose attribute is $c_i$ for every $i \in [r]$. Formally, the Vertex-Attribute-Constrained Densest $k$-Subgraph (VAC-D$k$S) Problem can be defined as follows.

**Definition 1** (Vertex-Attribute-Constrained Densest $k$-Subgraph (VAC-D$k$S) Problem). *Given a weighted, undirected, and simple graph $\mathcal{G} = (\mathcal{V}, \mathcal{E}, w)$ with at least one positive weight, a partition of $\mathcal{V}$ into $r$ subsets $\mathcal{C}_1, \mathcal{C}_2, \ldots, \mathcal{C}_r$ based on vertex attribute values,[2] and non-negative integers $k, k_1, k_2, \ldots, k_r$. VAC-DkS seeks a subset of $k$ vertices that includes at least $k_i$ vertices from each group $\mathcal{C}_i$ for every $i \in [r]$, and which maximizes the total edge weight (or the number of edges in the case of unweighted graphs) in the induced subgraph of $\mathcal{G}$. Without loss of generality, $1 \le k \le n$, $1 \le r \le n$, $0 \le k_i \le |\mathcal{C}_i|$, $\forall i \in [r]$, and $\sum_{i \in [r]} k_i \le k$.*

Let $\boldsymbol{x} \in \{0,1\}^n$ be an indicator vector of a subset of $\mathcal{V}$. VAC-D$k$S can be formulated as

$$\max_{\boldsymbol{x} \in \mathbb{R}^n} \quad f(\boldsymbol{x}) = \boldsymbol{x}^\top \boldsymbol{A} \boldsymbol{x}$$
$$\text{s.t.} \quad \boldsymbol{x} \in \mathcal{B}_k^n \cap \mathcal{F}, \tag{1}$$

---

[1] The assumption of at least one positive weight is required only for the counterexample constructed in Subsection 3.1. Consequently, only Corollary 2 relies on this constraint. The other theorems, lemmas, and Corollary 1 presented in Section 3 hold without this constraint.

[2] Throughout this paper, the term *group* refers to one of these subsets, i.e., the collection of vertices having the same attribute value.

where $\boldsymbol{A} \in \mathbb{R}^{n \times n}$ is the weighted adjacency matrix of $\mathcal{G}$, $\mathcal{B}_k^n = \{\boldsymbol{x} \in \mathbb{R}^n \mid \boldsymbol{x} \in \{0,1\}^n, \sum_{i \in [n]} x_i = k\}$, and $\mathcal{F} = \{\boldsymbol{x} \in \mathbb{R}^n \mid \sum_{i \in \mathcal{C}_j} x_i \geq k_j, \forall j \in [r]\}$.

Compared with the existing formulations in Anagnostopoulos et al. (2020; 2024); Miyauchi et al. (2023); Kariotakis et al. (2025), our formulation offers the following advantages:

- The formulations in Anagnostopoulos et al. (2020; 2024); Miyauchi et al. (2023); Kariotakis et al. (2025) do not allow explicit control of the subgraph size, with the result that they can extract large but loosely connected subgraphs.

- Compared with Anagnostopoulos et al. (2020; 2024) and Problem 1 in Miyauchi et al. (2023), which impose a common upper bound on the proportion of each vertex group in the solution, our formulation allows more flexible control over group composition tailored to end-user requirements by appropriate variation of the size parameters $\{k_i\}_{i=1}^r$ and $k$.

- Gajewar & Das Sarma (2012) and Problem 2 in Miyauchi et al. (2023) allow setting a lower bound on the number of vertices from each group in the solution, but without controlling the subgraph size, it cannot ensure a meaningful lower bound on group proportions. Our formulation addresses this by jointly constraining subgraph size and group representation.

- The formulations in Kariotakis et al. (2025) adjust group composition through regularization and support only a single attribute constraint. Furthermore, ensuring that the extracted subgraph satisfies a target group proportion requires tuning the regularization parameter via bisection-search, thereby increasing complexity and limiting applicability. In contrast, our formulation requires no parameter tuning (the size parameters are directly specified as problem input) and naturally handles multiple attribute constraints.

When considering ways to constrain subgraph size, at-least-$k$ and at-most-$k$ are two alternatives. However, both present notable drawbacks in our setting. At-least-$k$ constraints, similar to DSG-based formulations, tend to select large but loosely connected subgraphs, which cannot guarantee meaningful lower bounds on group proportions. Meanwhile, at-most-$k$ constraints may result in much smaller solutions, limiting their practical significance. Therefore, we focus on the exact-$k$ constraint, which allows us to precisely control the subgraph size and ensure meaningful group proportions.

## 3 Main Theoretical Results

**Theorem 1.** *VAC-DkS is NP-hard, and at least as difficult to approximate as DkS.*

*Proof.* D$k$S is a special case of VAC-D$k$S when $k_i = 0$, $\forall i \in [r]$. $\square$

Considering that D$k$S is provably difficult to approximate (Khot, 2006; Manurangsi, 2017) and the best known polynomial-time approximation algorithm for D$k$S can only achieve an $O(n^{1/4+\epsilon})$-approximation (Bhaskara et al., 2010), relaxing the combinatorial constraint in (1) to its convex hull and solving it through numerical optimization algorithms is a natural choice. Hence, we need to first find the convex hull of $\mathcal{B}_k^n \cap \mathcal{F}$.

**Theorem 2.** *The convex hull of $\mathcal{B}_k^n \cap \mathcal{F}$ is $\mathcal{D}_k^n \cap \mathcal{F}$, where $\mathcal{D}_k^n = \{\boldsymbol{x} \in \mathbb{R}^n \mid \boldsymbol{x} \in [0,1]^n, \sum_{i \in [n]} x_i = k\}$.*

*Proof.* Please refer to Appendix A. $\square$

Since $\mathcal{D}_k^n \cap \mathcal{F}$ is the convex hull of $\mathcal{B}_k^n \cap \mathcal{F}$, we relax (1) to the following continuous optimization problem

$$
\begin{aligned}
\max_{\boldsymbol{x} \in \mathbb{R}^n} \quad & f(\boldsymbol{x}) = \boldsymbol{x}^\mathsf{T} \boldsymbol{A} \boldsymbol{x} \\
\text{s.t.} \quad & \boldsymbol{x} \in \mathcal{D}_k^n \cap \mathcal{F}.
\end{aligned}
\tag{2}
$$

However, Corollary 2 in Lu et al. (2025a) shows that $\arg\max_{\boldsymbol{x}\in\mathcal{B}_k^n} f(\boldsymbol{x}) \subseteq \arg\max_{\boldsymbol{x}\in\mathcal{D}_k^n} f(\boldsymbol{x})$ does not hold in general for unweighted D$k$S. Since unweighted D$k$S is a special case of VAC-D$k$S, the relaxation from (1) to (2) is also not tight for VAC-D$k$S in general. In this paper, we adopt the definition of tightness of a relaxation from Lu et al. (2025a), i.e., a relaxation is tight if every optimal solution to the original problem remains optimal for the relaxed problem.

A technique known as *Diagonal Loading* has been used in Yuan & Zhang (2013); Barman (2018); Hager et al. (2016); Liu et al. (2024); Lu et al. (2025a) to either guarantee the tightness of D$k$S relaxation or improve the solution quality. The starting point is that for D$k$S, we may *equivalently* reformulate

$$\max_{\boldsymbol{x}\in\mathbb{R}^n} \quad f(\boldsymbol{x}) = \boldsymbol{x}^\top \boldsymbol{A}\boldsymbol{x}$$
$$\text{s.t.} \quad \boldsymbol{x} \in \mathcal{B}_k^n, \tag{3}$$

as

$$\max_{\boldsymbol{x}\in\mathbb{R}^n} \quad g(\boldsymbol{x}) = \boldsymbol{x}^\top (\boldsymbol{A} + \lambda \boldsymbol{I})\boldsymbol{x}$$
$$\text{s.t.} \quad \boldsymbol{x} \in \mathcal{B}_k^n, \tag{4}$$

where $\lambda$ is a non-negative diagonal loading parameter. Relaxing (4) yields

$$\max_{\boldsymbol{x}\in\mathbb{R}^n} \quad g(\boldsymbol{x}) = \boldsymbol{x}^\top (\boldsymbol{A} + \lambda \boldsymbol{I})\boldsymbol{x}$$
$$\text{s.t.} \quad \boldsymbol{x} \in \mathcal{D}_k^n. \tag{5}$$

Recently, Lu et al. (2025a) proved that $\arg\max_{\boldsymbol{x}\in\mathcal{B}_k^n} g(\boldsymbol{x}) \subseteq \arg\max_{\boldsymbol{x}\in\mathcal{D}_k^n} g(\boldsymbol{x})$ holds for every unweighted, undirected, and simple graph $\mathcal{G}$ and every $k$ if and only if the diagonal loading parameter $\lambda \geq 1$. Lu et al. (2025a) further showed the impact of $\lambda$ on the optimization landscape of (5), suggesting that a larger $\lambda$ can make the optimization landscape more challenging.

The motivation for adopting this diagonal loading technique is twofold: adding the $\lambda\|\boldsymbol{x}\|_2^2$ not only ensures the equivalence between the discrete problems (3) and (4), but also drives the continuous relaxation (5) towards integral solutions.

Similarly, (1) can be equivalently reformulated as

$$\max_{\boldsymbol{x}\in\mathbb{R}^n} \quad g(\boldsymbol{x}) = \boldsymbol{x}^\top (\boldsymbol{A} + \lambda \boldsymbol{I})\boldsymbol{x}$$
$$\text{s.t.} \quad \boldsymbol{x} \in \mathcal{B}_k^n \cap \mathcal{F}. \tag{6}$$

By relaxing (6), we obtain

$$\max_{\boldsymbol{x}\in\mathbb{R}^n} \quad g(\boldsymbol{x}) = \boldsymbol{x}^\top (\boldsymbol{A} + \lambda \boldsymbol{I})\boldsymbol{x}$$
$$\text{s.t.} \quad \boldsymbol{x} \in \mathcal{D}_k^n \cap \mathcal{F}. \tag{7}$$

To achieve higher-quality results in solving problem (7) using numerical optimization algorithms, we need to analyze the impact of the diagonal loading parameter $\lambda$ on the tightness of the relaxation from (6) to (7) and the optimization landscape of (7).

Note that if the relaxation from (6) to (7) is tight when $\lambda = \lambda^*$, where $\lambda^*$ represents the minimum value of the diagonal loading parameter to guarantee the tightness from (6) to (7), then the sets of optimal solutions of (6) and (7) are the same when $\lambda > \lambda^*$ because $\|\boldsymbol{x}\|_2^2 = k$ if and only if $\boldsymbol{x} \in \mathcal{B}_k^n \cap \mathcal{F}$, within the domain $\mathcal{D}_k^n \cap \mathcal{F}$. Therefore, we only need to derive the minimum value of the diagonal loading parameter to guarantee the tightness from (6) to (7).

## 3.1 Tightness of the Relaxation

Corollary 2 in Lu et al. (2025a) constructs counterexamples to establish that $\lambda = 1$ is a lower bound for the minimum value of the diagonal loading parameter to ensure the relaxation from (4) to (5) is tight for

unweighted graphs. For weighted graphs, consider a graph in which all edge weights are identical. In this case, since the structure is equivalent (up to a scaling) to an unweighted graph, we can apply the same construction to show that $\lambda = w_{\max}$ serves as a lower bound on the diagonal loading parameter to ensure tightness of the relaxation from (4) to (5), where $w_{\max}$ denotes the maximum edge weight in $\mathcal{G}$. Since D$k$S is a special case of VAC-D$k$S, this also implies that $\lambda = w_{\max}$ serves as a lower bound for the tightness of the relaxation from (6) to (7).

We next consider an upper bound on the minimum value of the diagonal loading parameter. There are two key challenges in deriving this upper bound: handling the attribute constraints introduced in VAC-D$k$S, and managing the complication introduced by edge weights.

**Theorem 3.** *Given any $\lambda \geq w_{\max}$ and a non-integral feasible $\boldsymbol{x}$ of (7), we can always find an integral feasible $\boldsymbol{x}'$ of (7) such that $g(\boldsymbol{x}') \geq g(\boldsymbol{x})$.*

*Proof.* Please refer to Appendix B. $\qquad\square$

**Corollary 1.** *If $\lambda \geq w_{\max}$, then there always exists an integral global maximizer of (7), which implies that the relaxation from (6) to (7) is tight when $\lambda \geq w_{\max}$.*

Corollary 1 shows that $\lambda = w_{\max}$ is an upper bound for the minimum value of the diagonal loading to ensure the tightness. Combining the previously obtained lower bound with this upper bound, we can derive the following corollary.

**Corollary 2.** *$\lambda = w_{\max}$ is the minimum value of the diagonal loading parameter to guarantee the tightness from (6) to (7).*

## 3.2 Landscape Analysis of the Relaxation

Having characterized the role of the diagonal loading parameter in ensuring tightness, we now examine its influence on the optimization landscape of (7).

**Lemma 1.** *There does not exist a non-integral local maximizer of (7) when $\lambda > w_{\max}$.*

*Proof.* Please refer to Appendix C. $\qquad\square$

**Theorem 4.** *Given $\lambda_2 > \lambda_1 > w_{\max}$, if $\boldsymbol{x}$ is a local maximizer of (7) with the diagonal loading parameter $\lambda_1$, then $\boldsymbol{x}$ is also a local maximizer of (7) with the diagonal loading parameter $\lambda_2$.*

*Proof.* Please refer to Appendix D. $\qquad\square$

In conclusion, Corollary 2 shows that $\lambda = w_{\max}$ is the minimum value of $\lambda$ to ensure the tightness from (6) to (7), while Theorem 4 shows that a larger $\lambda$ can make the optimization landscape more challenging. Through a more sophisticated rounding technique, Corollary 2 and Theorem 4 offer a significant and non-trivial extension of the results from the unweighted D$k$S problem to the more general weighted VAC-D$k$S problem, addressing both relaxation tightness and optimization landscape analysis.

## 4 Algorithms for VAC-D$k$S

Considering that VAC-D$k$S generalizes D$k$S, it is natural to attempt to generalize state-of-the-art algorithms developed for D$k$S to handle the more general VAC-D$k$S problem. In particular, L-ADMM (Konar & Sidiropoulos, 2021), Extreme Point Pursuit (EXPP) (Liu et al., 2024), the Frank–Wolfe algorithm (Lu et al., 2025a), and the parameterization approach (Lu et al., 2025a) have demonstrated strong performance on D$k$S in terms of solution quality and computational efficiency.

Projection-based algorithms are commonly employed for solving D$k$S (Hager et al., 2016; Liu et al., 2024). However, the projection onto the feasible set $\mathcal{D}_k^n \cap \mathcal{F}$ lacks a closed-form solution in VAC-D$k$S. The main culprit is the introduction of $r$ attribute constraints in VAC-D$k$S which complicates the computation of the

projection operator, rendering it computationally expensive and inefficient. Similarly, L-ADMM (Konar & Sidiropoulos, 2021) faces challenges when extended to VAC-D$k$S due to the additional $r$ variables introduced by attribute constraints, making the subproblems difficult to solve. Moreover, these constraints impede the straightforward generalization of the D$k$S parameterization approach (Lu et al., 2025a) to VAC-D$k$S. Consequently, we advocate for the Frank–Wolfe algorithm, a first-order, projection-free method, to efficiently tackle problem (7).

### 4.1 The Frank–Wolfe Algorithm

**Initialization:** Since problem (7) is non-convex, the choice of initialization can significantly affect the quality of the final solution. To this end, we use the procedure described in Algorithm 1, which constructs an initial feasible solution that satisfies the attribute constraints while distributing values as uniformly as possible. We empirically observed in our experiments that this is a good choice of initialization for problem (7).

**Algorithm:** The pseudo-code for the Frank–Wolfe algorithm is presented in Algorithm 2. Line 4 in Algorithm 2 computes the gradient. Lines 5 to 9 solve the linear maximization problem

$$\boldsymbol{s}^{(t)} \in \arg \max_{\boldsymbol{s} \in \mathcal{D}_k^n \cap \mathcal{F}} \boldsymbol{s}^{\mathsf{T}} \boldsymbol{g}^{(t)}. \tag{8}$$

While projection onto the constraint set $\mathcal{D}_k^n \cap \mathcal{F}$ is challenging, problem (8) admits a closed-form solution, making the Frank–Wolfe algorithm a natural and efficient choice for solving (7). Line 10 calculates the update direction, and Line 11 determines the step size. This step size rule guarantees convergence of the Frank–Wolfe algorithm to a stationary point of (7) (Bertsekas, 2016, p. 268), and experiments in Lu et al. (2025a) demonstrate that it converges faster than the scheme proposed by Lacoste-Julien (2016).

---

**Algorithm 1:** The initialization for Algorithm 2

**Input:** The subgraph size $k$, the sets of vertex attributes $\mathcal{C}_1, \mathcal{C}_2, \ldots, \mathcal{C}_r$, and the parameters for the attribute constraints $k_1, k_2, \ldots, k_r$.

**Initialization:** $\boldsymbol{x}$ is a zero vector of length $n$.

1 **for** $i = 1, 2, \ldots, r$ **do**
2 $\quad x[\mathcal{C}_i] \leftarrow \frac{k_i}{|\mathcal{C}_i|}$;
3 $residual \leftarrow k - \sum_{i \in [r]} k_i$;
4 **while** $residual > 0$ **do**
5 $\quad \mathcal{M} \leftarrow \{j \in [n] \mid x_j < 1\}$;
6 $\quad share \leftarrow \frac{residual}{|\mathcal{M}|}$;
7 $\quad$ **for** $j \in \mathcal{M}$ **do**
8 $\quad\quad update \leftarrow \min\{share, 1 - x_j\}$;
9 $\quad\quad x_j \leftarrow x_j + update$;
10 $\quad\quad residual \leftarrow residual - update$;
11 **return** $\boldsymbol{x}$

---

**Complexity Analysis:** To highlight the efficiency of our approach, we analyze the time complexity of the initialization step (Algorithm 1) and the per-iteration cost of the Frank–Wolfe algorithm (Algorithm 2).

For the initialization step, the for-loop in Lines 1 and 2 takes $O(n)$ time. Line 3 takes $O(r)$ time. For the while-loop in Lines 4 to 10, each iteration takes $O(n)$ time. Note that the entries corresponding to each group of vertices remain equal after each iteration, and the residual is greater than zero only if at least one entry reaches 1 in that iteration. Hence, if the residual is still positive after an iteration, at least one group's entries become 1, implying that the total number of iterations is at most $r$. Therefore, the total time complexity of Algorithm 1 is $O(rn)$. Note that in practice, the number of groups $r$ is usually much smaller than $n$, so the initialization step is typically efficient.

---

**Algorithm 2:** The Frank–Wolfe algorithm for (7)

---

**Input:** The weighted adjacency matrix $\boldsymbol{A}$, the subgraph size $k$, the sets of vertex attributes $\mathcal{C}_1, \mathcal{C}_2, \ldots, \mathcal{C}_r$, the parameters for the attribute constraints $k_1, k_2, \ldots, k_r$, and the diagonal loading parameter $\lambda$.

**Initialization:** $\boldsymbol{x}^{(1)}$ is a feasible point initialized by Algorithm 1, $\boldsymbol{s}^{(1)}, \boldsymbol{s}^{(2)}, \ldots$ are zero vectors of dimension $n$, and $\mathcal{H}^{(1)}, \mathcal{H}^{(2)}, \ldots$ are empty sets.

**1** $L \leftarrow \|\boldsymbol{A} + \lambda \boldsymbol{I}\|_2$;

**2** $k' \leftarrow \sum_{i \in [r]} k_i$;

**3 while** *the convergence criterion is not met* **do**

**4**     $\boldsymbol{g}^{(t)} \leftarrow (\boldsymbol{A} + \lambda \boldsymbol{I}) \boldsymbol{x}^{(t)}$;

**5**     **for** $i = 1, 2, \ldots, r$ **do**

**6**        $\boldsymbol{s}^{(t)}[\text{top}_{k_i}(\boldsymbol{g}^{(t)}, \mathcal{C}_i)] \leftarrow 1$;

**7**        $\mathcal{H}^{(t)} \leftarrow \mathcal{H}^{(t)} \cup \text{top}_{k_i}(\boldsymbol{g}^{(t)}, \mathcal{C}_i)$

**8**     **if** $k > k'$ **then**

**9**        $\boldsymbol{s}^{(t)}[\text{top}_{k-k'}(\boldsymbol{g}^{(t)}, [n] \backslash \mathcal{H}^{(t)})] \leftarrow 1$;

**10**     $\boldsymbol{d}^{(t)} \leftarrow \boldsymbol{s}^{(t)} - \boldsymbol{x}^{(t)}$;

**11**     $\gamma^{(t)} \leftarrow \min\left\{1, \frac{(\boldsymbol{g}^{(t)})^{\mathsf{T}} \boldsymbol{d}^{(t)}}{L \|\boldsymbol{d}^{(t)}\|_2^2}\right\}$;

**12**     $\boldsymbol{x}^{(t+1)} \leftarrow \boldsymbol{x}^{(t)} + \gamma^{(t)} \boldsymbol{d}^{(t)}$;

**13**     $t \leftarrow t + 1$;

---

For the Frank–Wolfe algorithm, if the Lipschitz constant is calculated by the Power method and treat the number of iterations for the Power method as a constant, then it takes $O(m + n)$ time to calculate the constant because there are $O(m + n)$ non-zeros elements in $\boldsymbol{A} + \lambda \boldsymbol{I}$. Similarly, Line 4 takes $O(m + n)$ time to calculate the gradient because $\boldsymbol{A} + \lambda \boldsymbol{I}$ has $O(m + n)$ non-zeros elements. For Lines 5 to 7, each inner iteration can be implemented by first building a max-heap in $O(|\mathcal{C}_i|)$ time using Floyd's algorithm, and then extracting the top-$k_i$ elements in $O(k_i \log |\mathcal{C}_i|)$ time. Summing over all $i \in [r]$, the total worst-case time complexity is $O(n + k \log n)$. Alternatively, if quickselect is used to find the top-$k_i$ elements in each group, the average time complexity per group reduces to $O(|\mathcal{C}_i|)$, resulting in an overall average-case complexity of $O(n)$. Similarly, for Lines 8 and 9, using a max-heap requires $O(n + k \log n)$ time in the worst case. Alternatively, using quickselect leads to an average-case complexity of $O(n)$. For Lines 10 to 13, it takes $O(n)$ time. Therefore, the per-iteration time complexity of Algorithm 2 is $O(m + n + k \log n)$ when using a heap-based implementation. Alternatively, an average-case complexity of $O(m + n)$ is achievable via quickselect.

The per-iteration complexity of Algorithm 2 matches that of its counterpart for the classical D$k$S problem, and is independent of the number of attribute constraints $r$, highlighting the efficiency and scalability of the Frank–Wolfe approach for solving the more general VAC-D$k$S problem.

## 4.2 Baseline Algorithms and Upper Bound for VAC-D$k$S

VAC-D$k$S is a new problem introduced in this paper, and there is no existing baseline in the literature to evaluate the effectiveness of the proposed Frank–Wolfe algorithm for solving (7). To address this, we draw inspiration from the Greedy Peeling algorithm (Asahiro et al., 2000; Charikar, 2000) and the low-rank bilinear optimization (LRBO) algorithm (Papailiopoulos et al., 2014), and generalize them to the VAC-D$k$S setting. Additionally, we derive an upper bound on the optimal edge weight to further assess solution quality.

### 4.2.1 The Greedy Peeling Algorithm

We first adapt the classical Greedy Peeling algorithm (Asahiro et al., 2000; Charikar, 2000) as a baseline for VAC-D$k$S. The original algorithm iteratively removes the vertex with the minimum (weighted) degree, breaking ties arbitrarily, until $k$ vertices remain. To satisfy the attribute constraints in VAC-D$k$S, we modify

the peeling criterion to ensure that the number of vertices from each attribute group remains above its respective threshold during the peeling process.

The time complexity of the Greedy Peeling algorithm depends on the data structure used to maintain node degrees. For unweighted graphs, a bucket queue can be used to achieve $O(m + n)$ time complexity. For weighted graphs, bucket-based methods no longer apply. Using Fibonacci heaps yields a total amortized time complexity of $O(m + n \log n)$.

### 4.2.2 The Low-Rank Bilinear Optimization (LRBO) Algorithm

The second algorithm is based on the LRBO approach proposed by Papailiopoulos et al. (2014). The LRBO approach with rank-$d$ approximation for D$k$S has a time complexity of $O(n^{d+1})$, making only the rank-1 approximation practically tractable for moderate-size problems. Therefore, we focus solely on the rank-1 case in this paper.

Let $\lambda_1$ and $\boldsymbol{v}_1$ be the largest eigenvalue (in magnitude) of $\boldsymbol{A}$ and the corresponding eigenvector of the largest eigenvalue, respectively. Let $\boldsymbol{A}_1 = \boldsymbol{v}_1\boldsymbol{u}_1^\top$, where $\boldsymbol{u}_1 = \lambda_1\boldsymbol{v}_1$. The rank-1 case solves the following problem:

$$\max_{\boldsymbol{x},\boldsymbol{y}\in\mathcal{B}_k^n\cap\mathcal{F}} \boldsymbol{x}^\top\boldsymbol{v}_1\boldsymbol{u}_1^\top\boldsymbol{y} = \max_{\boldsymbol{y}\in\mathcal{B}_k^n\cap\mathcal{F}} \left[\max_{\boldsymbol{x}\in\mathcal{B}_k^n\cap\mathcal{F}} \boldsymbol{x}^\top\boldsymbol{v}_y\right], \tag{9}$$

where $\boldsymbol{v}_y = c_1\boldsymbol{v}_1$ and $c_1 = \boldsymbol{u}_1^\top\boldsymbol{y}$.

For the subproblem $\max_{\boldsymbol{x}\in\mathcal{B}_k^n\cap\mathcal{F}} \boldsymbol{x}^\top\boldsymbol{v}_y$, we only need to consider the following two linear maximization problems $\max_{\boldsymbol{x}\in\mathcal{B}_k^n\cap\mathcal{F}} \boldsymbol{x}^\top\boldsymbol{v}_1$ and $\max_{\boldsymbol{x}\in\mathcal{B}_k^n\cap\mathcal{F}} -\boldsymbol{x}^\top\boldsymbol{v}_1$. After solving these two problems, $\boldsymbol{y}$ can be obtained by solving two other corresponding linear maximization problems.

LRBO requires computing the largest eigenvalue and corresponding eigenvector of $\boldsymbol{A}$, which takes $O(m + n)$ time. Besides that, LRBO also requires solving linear maximization subproblems similar to those in the Frank–Wolfe algorithm. As analyzed previously, the total time complexity of LRBO is $O(m + n + k \log n)$ when using a binary heap in the worst case, or $O(m+n)$ on average when using a quickselect-based approach.

### 4.2.3 An Upper Bound on the Edge Weight

To better interpret the quality of a solution, we define the normalized edge weight of a solution as

$$\text{Normalized Edge Weight} = \frac{\text{Total Edge Weight}}{w_{\max}\binom{k}{2}}. \tag{10}$$

We now generalize the upper bound on normalized edge density for the unweighted D$k$S problem proposed by Papailiopoulos et al. (2014) to an upper bound on the normalized edge weight for the weighted VAC-D$k$S problem in the following theorem.

**Theorem 5.** *The optimal normalized edge weight (or the normalized edge density in the case of unweighted graphs) of VAC-DkS can be bounded by*

$$\min\left\{1, \frac{\boldsymbol{x}^{*\top}\boldsymbol{A}_1\boldsymbol{y}^*}{w_{\max}k(k-1)} + \frac{\sigma_2(\boldsymbol{A})}{w_{\max}(k-1)}, \frac{\sigma_1(\boldsymbol{A})}{w_{\max}(k-1)}\right\}, \tag{11}$$

*where $(\boldsymbol{x}^*, \boldsymbol{y}^*)$ are an optimal solution to (9) and $\sigma_i(\boldsymbol{A})$ denotes the $i$-th largest singular value of $\boldsymbol{A}$.*

*Proof.* Please refer to Appendix E. □

## 5 Experimental Results

### 5.1 Datasets

We evaluate our method on both real-world attributed graphs and synthetic graphs. The real-world benchmarks include the following commonly used datasets:

Table 1: Statistics of real-world datasets ($n$ is the number of vertices, $m$ is the number of edges, and $r$ is the number of groups).

| Name | $n$ | $m$ | $r$ |
|---|---|---|---|
| Books | 92 | 374 | 2 |
| Blogs | 1,222 | 16,714 | 2 |
| Wikipedia | 11,631 | 170,773 | 2 |
| Twitter | 18,470 | 48,053 | 2 |
| GitHub | 37,700 | 289,003 | 2 |
| LastFM | 7,624 | 27,806 | 18 |

- **Political Books (Books)**: In this network, vertices represent books on United States politics, and edges represent co-purchasing relationships. Each vertex has an attribute indicating its political leaning. The network was downloaded from `https://github.com/SotirisTsioutsiouliklis/FairLaR` and the original network is also available at `https://websites.umich.edu/~mejn/netdata/`.

- **Political Blogs (Blogs)** (Adamic & Glance, 2005): In this network, vertices represent blogs on United States politics, and edges represent hyperlinks between them. Each vertex has an attribute indicating its political leaning. The network was downloaded from `https://github.com/SotirisTsioutsiouliklis/FairLaR`.

- **Wikipedia Crocodile (Wikipedia)** (Rozemberczki et al., 2021): In this network, vertices represent Wikipedia pages related to crocodiles, and edges represent mutual links between them. Each vertex has an attribute indicating whether it is popular. The network was downloaded from `https://github.com/benedekrozemberczki/FEATHER`.

- **Political Retweet (Twitter)** (Rossi & Ahmed, 2015): In this network, vertices represent Twitter users, and edges represent retweet relationships between them. Each vertex has an attribute indicating the user's political leaning. The network was downloaded from `https://github.com/SotirisTsioutsiouliklis/FairLaR`.

- **GitHub Developer (GitHub)** (Rozemberczki et al., 2021): In this network, vertices represent GitHub developers, and edges represent mutual follow relationships between them. Each vertex has an attribute indicating the developer's specialization in either machine learning or web development. The network was downloaded from `https://snap.stanford.edu/data/github-social.html`.

- **LastFM Asia (LastFM)** (Rozemberczki & Sarkar, 2020): In this network, vertices represent LastFM users in Asian countries, and edges represent mutual follow relationships between them. Each vertex has an attribute indicating the user's country. The network was downloaded from `https://github.com/benedekrozemberczki/FEATHER`.

Table 1 summarizes key statistics of the real-world datasets, including the number of vertices, edges, and attribute groups.

While the real-world datasets commonly used in prior work serve as useful benchmarks, they have certain limitations. In particular, most are relatively small in scale, contain only unweighted edges, and involve binary group attributes. To enable evaluation on larger datasets, some approaches assign random attribute values to existing real-world graphs. However, this practice may weaken the natural correlation between attributes and graph structure, potentially reducing the effectiveness of the evaluation.

To address these limitations, we design a series of synthetic graphs based on the planted clique model. Specifically, we generate an Erdős–Rényi random graph $G(n, p)$ with $n$ vertices, where each edge is included independently with probability $p$. Each vertex is randomly assigned to one of $r$ groups with equal probability. We then plant a clique of size $k$ by selecting exactly $k/r$ vertices from each group, where $k$ is chosen to be divisible by $r$ to ensure equal allocation. This planted clique serves as the ground-truth dense community

for evaluating algorithm performance under multi-group settings. In the experimental results presented on these synthetic graphs, "success count" refers to the number of trials where an algorithm exactly recovers this planted ground-truth subgraph.

For unweighted graphs, edges are either present or absent according to this process. For weighted graphs, we assign edge weights differently: each edge in the initial Erdős–Rényi graph is given a weight sampled uniformly from the interval $[0.8, 1]$, and edges within the planted clique are set to 1.

This setup enables systematic evaluation of algorithm performance on weighted or unweighted graphs of various scales, with multiple groups and controlled attribute and structural properties.

## 5.2 Baselines and Implementation Details

We evaluate our method against the two baselines that we derived by generalizing their D$k$S version in Section 4.2: the (generalized) Greedy Peeling algorithm and the LRBO approach. In addition, we include a hybrid variant that initializes Algorithm 2 with the output of Greedy Peeling. We include this variant to test whether initializing with a high-quality heuristic output allows our method to achieve better results than starting from scratch.

All experiments were conducted on a workstation with an AMD Ryzen Threadripper 3970X CPU, 256 GB RAM, running Ubuntu 20.04. The implementation was done in Python 3.11.

For Frank–Wolfe, based on the tightness analysis in Corollary 2 and the landscape analysis in Theorem 4, we set the diagonal loading parameter to $w_{\max}$ (or 1 for unweighted graphs). The maximum iteration count is set to 500, which suffices for convergence in most real-world cases.

For Greedy Peeling, we use different implementations depending on the graph type. For unweighted graphs, we adopt a bucket queue for efficiency. For weighted graphs, we use a Fibonacci heap to maintain the peeling order. The Fibonacci heap is implemented via the `fibonacci-heap-mod` Python package.

For synthetic datasets, to ensure reproducibility, we fix the random seed for each of the $t$ repeated trials to values from 0 to $t-1$. This guarantees that all experiments are deterministic and results can be consistently reproduced.

We measure execution time by running each algorithm in a separate process to ensure memory isolation. Within each process, we perform a warm-up run using the same configuration and input graph to eliminate one-time initialization effects. The warm-up run is not timed; it is followed by a separate execution for measurement.

## 5.3 Binary-Attribute Graphs with Attribute-Constrained Groups

We consider multiple binary-attribute real-world graphs in Table 1, where each graph contains exactly one attribute-constrained group. We designate group 1 as the attribute-constrained group following the attribute convention used in `https://github.com/SotirisTsioutsiouliklis/FairLaR`. For each graph, at least $\lceil k \cdot \alpha \rceil$ vertices are selected from the attribute-constrained group, where $\alpha$ denotes the attribute-constrained group ratio.

Figure 1 and Figures 3 to 6 in Appendix F demonstrate that Frank–Wolfe with uniform initialization typically produces subgraphs with the highest density in most cases. Using the Greedy Peeling result as initialization for Frank–Wolfe yields density values higher than Greedy Peeling alone, and in some cases even achieves the highest subgraph density overall.

This strong empirical performance of Frank–Wolfe can be attributed to the theoretically grounded nature of our formulation. Section 3 establishes a relatively benign optimization landscape for our formulation (7) with appropriate diagonal loading, which makes it easier for our algorithm to find a high-quality solution.

Notably, Frank–Wolfe with uniform initialization exhibits a distinctive ability to discover subgraphs with imbalanced attribute composition but exceptionally high edge density. This property is particularly valuable in community discovery, as it helps uncover hidden, tightly connected attribute-constrained groups.

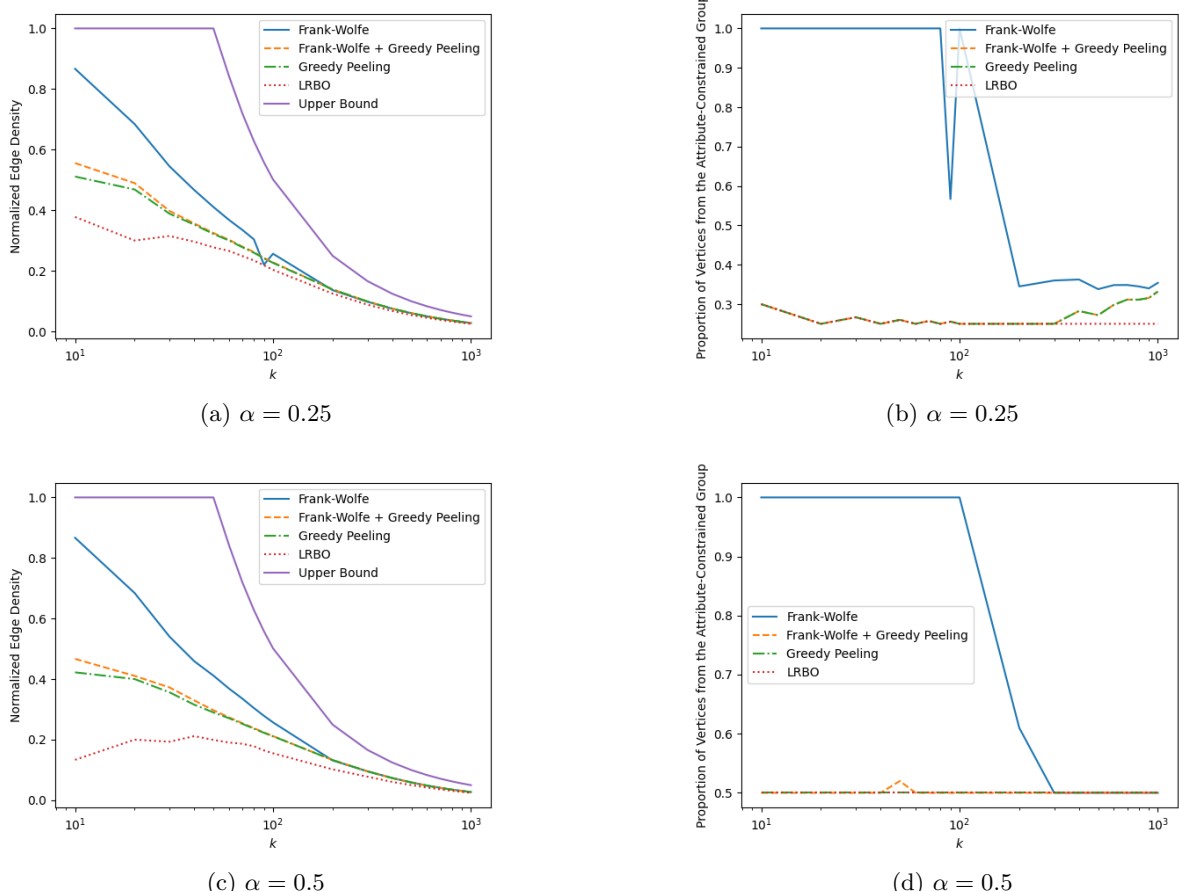

Figure 1: Normalized edge density and attribute-constrained group proportion on the Twitter dataset under different $\alpha$. Result: our Frank–Wolfe with uniform initialization (blue) outperforms other methods in terms of density for small $k$ and all methods have similar performance for large $k$.

### 5.4 Multi-Attribute-Value Graphs with Group Representation Constraints

We first evaluate our methods on the LastFM real-world dataset, which contains 18 attribute groups. We impose group representation constraints by requiring at least 5 or 10 vertices from each group in the extracted subgraphs. Figure 2 shows that all algorithms perform similarly, with the exception of LRBO, which exhibits worse results.

To further challenge these algorithms, we generate unweighted synthetic planted clique graphs with 3 attribute groups and significant background noise. All graphs share the same parameters—number of nodes $n = 10,000$, edge probability $p = 0.05$, and planted clique size $k = 30$—while varying only in random seeds to ensure reproducibility. We impose group representation constraints by requiring at least 5 vertices from each group in the extracted subgraphs.

Table 2 presents the performance of different algorithms on synthetic planted clique graphs with significant background noise and three attribute groups. Both Greedy Peeling and LRBO failed to recover the planted subgraph in any of the 20 runs. While Greedy Peeling achieved relatively high and stable normalized edge density $(0.823 \pm 0.109)$, it consistently selected dense regions formed by noisy connections, suggesting that the background noise effectively masked the true clique. In this sense, Greedy Peeling often found near-optimal solutions in terms of density but lacked the resolution to distinguish the planted structure from spurious dense subgraphs. Frank–Wolfe with uniform initialization recovered the planted subgraph in 13

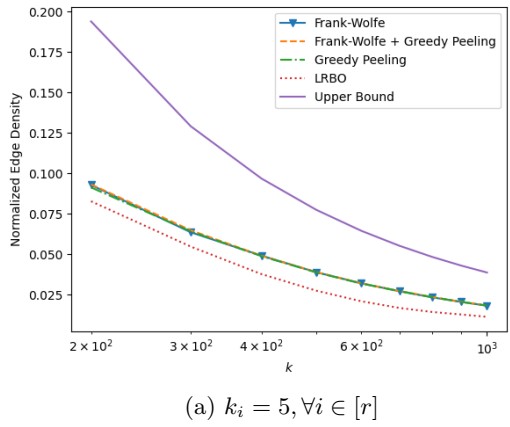 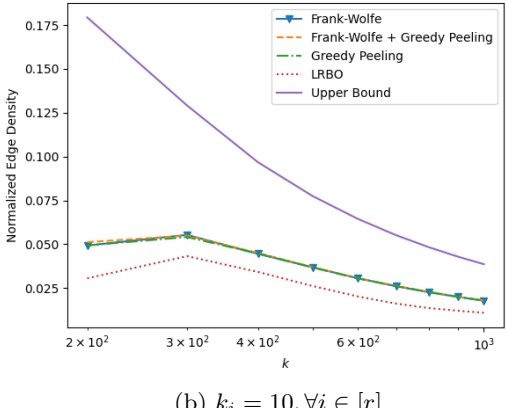

(a) $k_i = 5, \forall i \in [r]$          (b) $k_i = 10, \forall i \in [r]$

Figure 2: Normalized edge density on the LastFM dataset with 18 groups under different $k_i$. Result: LRBO (red) is outperformed by other methods in terms of density.

Table 2: Normalized edge density and success count for different algorithms on unweighted synthetic planted clique graphs ($n = 10,000$, $p = 0.05$, and $k = 30$) with 3 attribute groups. Normalized edge density values are reported as mean $\pm$ sample standard deviation over 20 runs.

| Algorithm | Normalized Edge Density | Success Count |
|---|---|---|
| LRBO | $0.074 \pm 0.011$ | 0 / 20 |
| Greedy Peeling | $0.823 \pm 0.109$ | 0 / 20 |
| Frank–Wolfe | $0.741 \pm 0.362$ | 13 / 20 |
| Frank–Wolfe + Greedy Peeling | $1.000 \pm 0.000$ | 20 / 20 |

out of 20 runs, reflecting its capacity to escape poor local optima, albeit with high variance ($0.741 \pm 0.362$). LRBO performed only marginally better than random, with very low density and no successful recoveries, demonstrating the poor performance of the spectral-based approach in noisy settings. When initialized with Greedy Peeling, Frank–Wolfe succeeded in all runs and achieved perfect density, further highlighting that a good heuristic starting point, though insufficient on its own, can be effectively refined by optimization. This result underscores the effectiveness of our proposed problem reformulation, which enables Frank–Wolfe to meaningfully navigate the solution space and recover meaningful structures even under significant noise.

## 5.5 Scalability on Large Unweighted and Weighted Graphs

To evaluate scalability in large-scale settings, we generate unweighted and weighted synthetic planted clique graphs with 3 attribute groups. All graphs share the same parameters—number of nodes $n = 200,000$, edge probability $p = 0.0025$, and planted clique size $k = 60$. We impose group representation constraints by requiring at least 10 vertices from each group in the extracted subgraphs.

Tables 3 and 4 show the results on large-scale unweighted and weighted planted clique graphs, respectively. While both Frank–Wolfe and Greedy Peeling successfully identify the planted structure in all runs, LRBO fails consistently across both settings. In terms of execution time, Frank–Wolfe outperforms Greedy Peeling, especially in the weighted setting where Greedy Peeling requires a Fibonacci heap to maintain correct peeling order, resulting in over $5\times$ longer runtimes. This speedup may be attributed to Frank–Wolfe's better cache locality and its algorithmic structure, which is more amenable to vectorization and parallelization. Our Frank–Wolfe implementation relies on `scipy`'s single-threaded sparse matrix-vector multiplication; employing parallelized libraries could yield further performance improvements, particularly for larger graphs.

Table 3: Normalized edge density, success count, and execution time for different algorithms on unweighted synthetic planted clique graphs ($n = 200,000$, $p = 0.0025$, and $k = 60$) with 3 attribute groups. Normalized edge density values and execution times are reported as mean $\pm$ sample standard deviation over 5 runs.

| Algorithm | Normalized Edge Density | Success Count | Execution Time (s) |
|---|---|---|---|
| LRBO | $0.094 \pm 0.036$ | 0 / 5 | $171.3 \pm 1.3$ |
| Greedy Peeling | $1.000 \pm 0.000$ | 5 / 5 | $226.1 \pm 0.5$ |
| Frank–Wolfe | $1.000 \pm 0.000$ | 5 / 5 | $185.6 \pm 0.8$ |

Table 4: Normalized edge weight, success count, and execution time for different algorithms on weighted synthetic planted clique graphs ($n = 200,000$, $p = 0.0025$, and $k = 60$) with 3 attribute groups. Normalized edge weight values and execution times are reported as mean $\pm$ sample standard deviation over 5 runs.

| Algorithm | Normalized Edge Weight | Success Count | Execution Time (s) |
|---|---|---|---|
| LRBO | $0.162 \pm 0.032$ | 0 / 5 | $173.6 \pm 3.9$ |
| Greedy Peeling | $1.000 \pm 0.000$ | 5 / 5 | $973.6 \pm 7.5$ |
| Frank–Wolfe | $1.000 \pm 0.000$ | 5 / 5 | $185.6 \pm 3.3$ |

### 5.6 Case Study: Greek Politics

We next conduct a case study on a dataset related to Greek politics (Stamatelatos et al., 2020), which was previously used by Fazzone et al. (2022) to analyze political divisions. The raw data is a weighted undirected graph consisting of 186 vertices and 17,185 edges, where the vertices represent Twitter accounts of Greek MPs (Members of Parliament) and Greek media outlets, and the edge weights indicate the audience similarity between two Twitter accounts. The network was downloaded from `https://github.com/tlancian/dith`. We manually labeled each vertex with a political orientation, where 0 denotes left-wing leaning and 1 denotes right-wing leaning. The final distribution consists of 95 vertices labeled 0 and 91 vertices labeled 1.

We set $k = 20$ and compared two cases: $k_1 = k_2 = 0$, which corresponds to the D$k$S problem, and $k_1 = k_2 = 10$, which corresponds to the perfectly balanced VAC-D$k$S problem. We used Algorithm 2 to solve these two problems.

Table 5 presents the subgraphs identified by the classical D$k$S algorithm and the proposed perfectly balanced VAC-D$k$S variant on the Greek Politics dataset. Interestingly, despite the added attribute constraints, the perfectly balanced VAC-D$k$S achieved a slightly higher normalized edge weight (0.391 vs. 0.376). This improvement can be attributed to the enhanced initialization provided by the attribute constraints, which help guide the algorithm away from suboptimal local solutions. In contrast to the perfectly balanced VAC-D$k$S, the classical D$k$S result is notably imbalanced: 80% of the selected vertices belong to the right-wing group, indicating a skewed extraction. It is also worth noting that both algorithms exclusively selected politicians and no media accounts. This is likely due to the broader and more mixed audience base of media outlets, which implies sparser audience connections with other accounts and thus lower pairwise similarity scores.

The classical D$k$S formulation tends to select center-right politicians with relatively cohesive and moderate ideological positions. The few left-wing politicians drawn in the mix are center-left, known for their relatively moderate demeanor.

The perfectly balanced VAC-D$k$S, on the other hand, pulls in a more politically heterogeneous mix that includes several individuals advocating less moderate views which are often prominently featured in public media and political discourse. Makis Voridis is a prominent representative of very right-wing views (Smith, 2011); Andreas Loverdos is known for his strong public stances on various policy issues; and Nikos Dendias is likewise known for his firm stance on defense and national priorities, including security. Interestingly, the VAC-D$k$S solution also includes several prominent figures that have toned down and moderated the political discourse—such as socialist leader Fofi Gennimata, and George Katrougalos who helped forge a treaty that

Table 5: Greek MPs and subgraph normalized edge weight extracted from the Greek Politics dataset by Algorithm 2. Labels indicate political leanings (**(0)**: left leaning, **(1)**: right leaning).

| D$k$S | Perfectly Balanced VAC-D$k$S |
|---|---|
| Fotini Arampatzi (1) | Vassilis Kikilias (1) |
| Evi Christofilopoulou (0) | Spyros Lykoudis (0) |
| Simos Kedikoglou (1) | Evi Christofilopoulou (0) |
| Odysseas Konstantinopoulos (0) | Odysseas Konstantinopoulos (0) |
| Kostas Skandalidis (0) | Kostas Skandalidis (0) |
| Gerasimos Giakoumatos (1) | Nikos Dendias (1) |
| Niki Kerameus (1) | Andreas Loverdos (0) |
| Giannis Kefalogiannis (1) | Miltiadis Varvitsiotis (1) |
| Miltiadis Varvitsiotis (1) | Notis Mitarachi (1) |
| Notis Mitarachi (1) | Makis Voridis (1) |
| Kostas Skrekas (1) | Giorgos Koumoutsakos (1) |
| Giorgos Koumoutsakos (1) | Christos Staikouras (1) |
| Christos Staikouras (1) | Olga Kefalogianni (1) |
| Giannis Plakiotakis (1) | George Katrougalos (0) |
| Theodoros Karaoglou (1) | Anna Asimakopoulou (1) |
| Anna Karamanli (1) | Markos Bolaris (0) |
| Stavros Kalafatis (1) | Elena Kountoura (0) |
| Anna Asimakopoulou (1) | Fofi Gennimata (0) |
| Nikitas Kaklamanis (1) | Nikitas Kaklamanis (1) |
| Yannis Maniatis (0) | Yannis Maniatis (0) |
| Normalized Edge Weight: 0.376 | Normalized Edge Weight: 0.391 |

was politically sensitive and contentious. Overall, the VAC-D$k$S solution is much more interesting than the D$k$S one. These results suggest that the vertex-attribute-constrained formulation is not only more balanced in representation but also more effective at highlighting structurally dense, cross-cutting subgraphs that reflect real-world political salience.

To further demonstrate the advantages of our D$k$S-based approach, we compare VAC-D$k$S with approaches based on alternative frameworks.

First, we apply the DFSG algorithm (using Greedy Peeling for the initial densest subgraph step), which is a DSG-based method proposed by Anagnostopoulos et al. (2020), to the 186-vertex Greek Politics dataset, which yields a subgraph containing 150 vertices—over 80% of the entire graph—with a normalized edge weight of only 0.112. This outcome empirically confirms the limitations of DSG-based formulations for tasks requiring the identification of specific, cohesive core communities: they lack effective size control and tend to produce large but loosely connected subgraph. This highlights the necessity of our D$k$S-based VAC-D$k$S framework.

Next, we apply the LFPR$_N$ algorithm with the allocation parameter $\phi = 0.5$, which is a centrality-based method proposed by Tsioutsiouliklis et al. (2021), to the Greek Politics dataset. Since LFPR$_N$ is designed for unweighted graphs, we first converted our weighted graph to an unweighted one using a threshold of 0.2. The output of LFPR$_N$ is shown in Table 6 in Appendix F. The LFPR$_N$ Top-20 ranking includes 6 media outlets, whereas our VAC-D$k$S identified a cohesive subgraph consisting exclusively of politicians. This difference highlights the fundamental difference between centrality-based methods (identifying influential entities) and our density-based approach (finding cohesive communities). Furthermore, the subset of politicians with the highest ranks identified by LFPR$_N$ is highly imbalanced: the 7 highest-ranked politicians are all from the PASOK party, unlike our VAC-DkS which can guarantee perfect balance when appropriately parameterized ($k_1 = k_2 = k/2$) due to its hard constraints.

## 6 Conclusion

In this paper, we introduced the Vertex-Attribute-Constrained Densest $k$-Subgraph (VAC-D$k$S) problem, a generalization of the classical D$k$S that incorporates vertex-attribute constraints. We showed that VAC-D$k$S is NP-hard, as it subsumes D$k$S as a special case. To address this challenge, we proposed an equivalent reformulation of VAC-D$k$S using diagonal loading, followed by a relaxation of the combinatorial constraint to its convex hull. Crucially, we proved that the relaxation is tight in general if and only if the diagonal loading parameter $\lambda \geq w_{\max}$, and provided landscape analysis to illustrate how $\lambda$ affects solution quality. We then designed a projection-free Frank–Wolfe algorithm to solve the relaxed problem efficiently. Extensive experiments demonstrate that our method achieves high-quality solutions across various settings and scales well to large graphs. Additionally, we illustrate an application of our method to a real-world political network in Greece. Our algorithm identifies a subgraph with balanced representation from both political camps, while still capturing individuals with strong ideological identities—an effect not observed with the classical D$k$S formulation. This case study highlights the practical relevance of attribute constraints in uncovering more representative and interpretable structures in real networks.

## 7 Acknowledgments

Aritra Konar was supported by the KU Leuven Special Research Fund (BOF/STG-22-040).

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

## A  Proof of Theorem 2

*Proof.* The Krein-Milman theorem (Rudin, 1991, Theorem 3.23) states that a non-empty, compact convex set is the closed convex hull of the set of its extreme points. Similar to Liu et al. (2024), we need to prove that a point is an extreme point of $\mathcal{D}_k^n \cap \mathcal{F}$ if and only if it is a point in $\mathcal{B}_k^n \cap \mathcal{F}$.

We first prove that a point is an extreme point of $\mathcal{D}_k^n \cap \mathcal{F}$ if it is a point in $\mathcal{B}_k^n \cap \mathcal{F}$. For any $\boldsymbol{x} \in \mathcal{B}_k^n \cap \mathcal{F}$, we have $\|\boldsymbol{x}\|_2^2 = k$. If $\boldsymbol{x}$ is not an extreme point of $\mathcal{D}_k^n \cap \mathcal{F}$, then there exists $\boldsymbol{y}, \boldsymbol{z} \in \mathcal{D}_k^n \cap \mathcal{F}$ and $\theta \in (0,1)$, such that $\boldsymbol{x} = \theta\boldsymbol{y} + (1-\theta)\boldsymbol{z}$. Since $\|\cdot\|_2^2$ is strictly convex, we can use the Jensen's inequality to derive the following contradiction:

$$k = \|\boldsymbol{x}\|_2^2 < \theta\|\boldsymbol{y}\|_2^2 + (1-\theta)\|\boldsymbol{z}\|_2^2 \le k. \tag{12}$$

Therefore, if a point is in $\mathcal{B}_k^n \cap \mathcal{F}$, then it is an extreme point of $\mathcal{D}_k^n \cap \mathcal{F}$.

Next, we prove that a point is an extreme point of $\mathcal{D}_k^n \cap \mathcal{F}$ only if it is a point in $\mathcal{B}_k^n \cap \mathcal{F}$. Suppose that $\boldsymbol{x} \in (\mathcal{D}_k^n \cap \mathcal{F})\backslash(\mathcal{B}_k^n \cap \mathcal{F})$. Let $\mathcal{M}(\boldsymbol{x}) = \{i \in [n] \mid 0 < x_i < 1\}$ and $\mathcal{M}_i(\boldsymbol{x}) = \{j \in \mathcal{C}_i \mid 0 < x_j < 1\}, \forall i \in [r]$. Since $\boldsymbol{x} \in (\mathcal{D}_k^n \cap \mathcal{F})\backslash(\mathcal{B}_k^n \cap \mathcal{F})$, we know that $|\mathcal{M}(\boldsymbol{x})| \ge 2$. We consider the following two cases:

- If there exists $i \in [r]$, such that $|\mathcal{M}_i(\boldsymbol{x})| \ge 2$, then we can find two distinct vertices $j, l \in \mathcal{M}_i(\boldsymbol{x})$. Let $\delta = \min\{x_j, x_l, 1-x_j, 1-x_l\}$, then we have $\boldsymbol{y} = \boldsymbol{x}+\delta(\boldsymbol{e}_j-\boldsymbol{e}_l) \in \mathcal{D}_k^n \cap \mathcal{F}$ and $\boldsymbol{z} = \boldsymbol{x}+\delta(\boldsymbol{e}_l-\boldsymbol{e}_j) \in \mathcal{D}_k^n \cap \mathcal{F}$, where $\boldsymbol{e}_j$ is the $j$-th vector of the canonical basis for $\mathbb{R}^n$. Since $\boldsymbol{x} = \frac{1}{2}\boldsymbol{y} + \frac{1}{2}\boldsymbol{z}$, we know that $\boldsymbol{x}$ is not an extreme point of $\mathcal{D}_k^n \cap \mathcal{F}$.

- If there does not exist $i \in [r]$, such that $|\mathcal{M}_i(\boldsymbol{x})| \geq 2$, then we can find two distinct set $\mathcal{M}_i(\boldsymbol{x})$ and $\mathcal{M}_j(\boldsymbol{x})$, such that $|\mathcal{M}_i(\boldsymbol{x})| = |\mathcal{M}_j(\boldsymbol{x})| = 1$. Since $\boldsymbol{x} \in [0,1]^n$ and $|\mathcal{M}_i(\boldsymbol{x})| \leq 1$, $\forall i \in [r]$, we have $\sum_{l \in \mathcal{C}_i \setminus \mathcal{M}_i(\boldsymbol{x})} x_l \geq k_i$ and $\sum_{l \in \mathcal{C}_j \setminus \mathcal{M}_j(\boldsymbol{x})} x_l \geq k_j$. Suppose that $l \in \mathcal{M}_i(\boldsymbol{x})$ and $q \in \mathcal{M}_j(\boldsymbol{x})$. Let $\delta = \min\{x_l, x_q, 1 - x_l, 1 - x_q\}$, then we have $\boldsymbol{y} = \boldsymbol{x} + \delta(\boldsymbol{e}_l - \boldsymbol{e}_q) \in \mathcal{D}_k^n \cap \mathcal{F}$ and $\boldsymbol{z} = \boldsymbol{x} + \delta(\boldsymbol{e}_q - \boldsymbol{e}_l) \in \mathcal{D}_k^n \cap \mathcal{F}$. Since $\boldsymbol{x} = \frac{1}{2}\boldsymbol{y} + \frac{1}{2}\boldsymbol{z}$, we know that $\boldsymbol{x}$ is not an extreme point of $\mathcal{D}_k^n \cap \mathcal{F}$.

Therefore, we can conclude that a point is an extreme point of $\mathcal{D}_k^n \cap \mathcal{F}$ only if it is a point in $\mathcal{B}_k^n \cap \mathcal{F}$. $\quad\square$

## B   Proof of Theorem 3

*Proof.* Let $\mathcal{M}(\boldsymbol{x}) = \{i \in [n] \mid 0 < x_i < 1\}$ and $\mathcal{M}_i(\boldsymbol{x}) = \{j \in \mathcal{C}_i \mid 0 < x_j < 1\}$, $\forall i \in [r]$. Since $\boldsymbol{x}$ is non-integral, we have $|\mathcal{M}(\boldsymbol{x})| = \sum_{i \in [r]} |\mathcal{M}_i(\boldsymbol{x})| \geq 2$.

If there exists $i \in [r]$, such that $|\mathcal{M}_i(\boldsymbol{x})| \geq 2$, then we can always find two distinct vertices $j, l \in \mathcal{M}_i(\boldsymbol{x})$ such that $\lambda x_j + s_j \geq \lambda x_l + s_l$, where $s_j = \sum_{q \in [n]} a_{jq} x_q$, $\forall j \in [n]$. Let $\delta = \min\{x_l, 1 - x_j\}$, $\boldsymbol{d} = \boldsymbol{e}_j - \boldsymbol{e}_l$, where $\boldsymbol{e}_j$ is the $j$-th vector of the canonical basis for $\mathbb{R}^n$, and $\hat{\boldsymbol{x}} = \boldsymbol{x} + \delta \boldsymbol{d}$. $\hat{\boldsymbol{x}}$ is still a feasible point of (7). To analyze the effect of the update on the objective function $g$, we consider the difference:

$$
\begin{aligned}
& g(\hat{\boldsymbol{x}}) - g(\boldsymbol{x}) \\
=& 2(x_j + \delta)(s_j - a_{jl}x_l) + \lambda(x_j + \delta)^2 + 2(x_l - \delta)(s_l - a_{jl}x_j) + \lambda(x_l - \delta)^2 + 2a_{jl}(x_j + \delta)(x_l - \delta) \\
& - 2x_j(s_j - a_{jl}x_l) - \lambda x_j^2 - 2x_l(s_l - a_{jl}x_j) - \lambda x_l^2 - 2a_{jl}x_j x_l \\
=& 2\delta(\lambda x_j + s_j - \lambda x_l - s_l) + 2(\lambda - a_{jl})\delta^2 \\
\geq& 0.
\end{aligned}
\tag{13}
$$

Hence, after the above update, the objective value $g(\hat{\boldsymbol{x}})$ is greater than or equal to the objective value $g(\boldsymbol{x})$ and the cardinality $|\mathcal{M}_i(\hat{\boldsymbol{x}})|$ is strictly smaller than the cardinality $|\mathcal{M}_i(\boldsymbol{x})|$. Repeat this update until the cardinality $|\mathcal{M}_i(\hat{\boldsymbol{x}})|$ is either 0 or 1.

After the aforementioned iteration, there is at most one non-integral entry in the indicator vector $\hat{\boldsymbol{x}}$ corresponding to the $i$-th group. Apply the same iteration to other groups until $|\mathcal{M}_i(\hat{\boldsymbol{x}})| \leq 1$, $\forall i \in [r]$.

After these iterations, if $|\mathcal{M}(\hat{\boldsymbol{x}})| = 0$, then we already have an integral feasible $\hat{\boldsymbol{x}}$ of (7) such that $g(\hat{\boldsymbol{x}}) \geq g(\boldsymbol{x})$. If $\hat{\boldsymbol{x}}$ is non-integral, then $|\mathcal{M}(\hat{\boldsymbol{x}})| \geq 2$, which implies that we can always find two distinct vertices $j, l \in \mathcal{M}(\hat{\boldsymbol{x}})$ such that $\lambda \hat{x}_j + \hat{s}_j \geq \lambda \hat{x}_l + \hat{s}_l$, where $\hat{s}_j = \sum_{q \in [n]} a_{jq} \hat{x}_q$, $\forall j \in [n]$. Let $\hat{\delta} = \min\{\hat{x}_l, 1 - \hat{x}_j\}$ and $\hat{\boldsymbol{d}} = \boldsymbol{e}_j - \boldsymbol{e}_l$. Since $\hat{\boldsymbol{x}} \in [0,1]^n$ and $|\mathcal{M}_i(\hat{\boldsymbol{x}})| \leq 1$, $\forall i \in [r]$, we have $\sum_{l \in \mathcal{C}_i \setminus \mathcal{M}_i(\hat{\boldsymbol{x}})} \hat{x}_l \geq k_i$, $\forall i \in [r]$, which implies that $\hat{\boldsymbol{x}} + \hat{\delta}\hat{\boldsymbol{d}}$ is feasible of (7). Similar to (13), we have the objective value $g(\hat{\boldsymbol{x}} + \hat{\delta}\hat{\boldsymbol{d}})$ is greater than or equal to the objective value $g(\hat{\boldsymbol{x}})$ and the cardinality $|\mathcal{M}(\hat{\boldsymbol{x}} + \hat{\delta}\hat{\boldsymbol{d}})|$ is strictly smaller than the cardinality $|\mathcal{M}(\hat{\boldsymbol{x}})|$. Repeat this update until the cardinality $|\mathcal{M}(\hat{\boldsymbol{x}})|$ is 0, then we obtain an integral feasible $\hat{\boldsymbol{x}}$ of (7) such that $g(\hat{\boldsymbol{x}}) \geq g(\boldsymbol{x})$. $\quad\square$

## C   Proof of Lemma 1

*Proof.* Let $\mathcal{M}(\boldsymbol{x}) = \{i \in [n] \mid 0 < x_i < 1\}$ and $\mathcal{M}_i(\boldsymbol{x}) = \{j \in \mathcal{C}_i \mid 0 < x_j < 1\}$, $\forall i \in [r]$. Since $\boldsymbol{x}$ is non-integral, we have $|\mathcal{M}(\boldsymbol{x})| = \sum_{i \in [r]} |\mathcal{M}_i(\boldsymbol{x})| \geq 2$. Considering the following two cases:

- If there exists $i \in [r]$, such that $|\mathcal{M}_i(\boldsymbol{x})| \geq 2$, then we can always find two distinct vertices $j, l \in \mathcal{M}_i(\boldsymbol{x})$ such that $\lambda x_j + s_j \geq \lambda x_l + s_l$, where $s_j = \sum_{q \in [n]} a_{jq} x_q$, $\forall j \in [n]$. Let $\hat{\delta} = \min\{x_l, 1 - x_j\}$, $\boldsymbol{d} = \boldsymbol{e}_j - \boldsymbol{e}_l$, where $\boldsymbol{e}_j$ is the $j$-th vector of the canonical basis for $\mathbb{R}^n$. For every $\delta \in (0, \hat{\delta}]$, since

$x + \delta d$ is still feasible of (7) and

$$
\begin{aligned}
&g(\boldsymbol{x} + \delta\boldsymbol{d}) - g(\boldsymbol{x}) \\
=&2(x_j + \delta)(s_j - a_{jl}x_l) + \lambda(x_j + \delta)^2 + 2(x_l - \delta)(s_l - a_{jl}x_j) + \lambda(x_l - \delta)^2 + 2a_{jl}(x_j + \delta)(x_l - \delta) \\
&- 2x_j(s_j - a_{jl}x_l) - \lambda x_j^2 - 2x_l(s_l - a_{jl}x_j) - \lambda x_l^2 - 2a_{jl}x_j x_l \\
=&2\delta(\lambda x_j + s_j - \lambda x_l - s_l) + 2(\lambda - a_{jl})\delta^2 \\
>&0,
\end{aligned}
\tag{14}
$$

we know that $\boldsymbol{d}$ is an ascent direction at $\boldsymbol{x}$.

- If there does not exist $i \in [r]$, such that $|\mathcal{M}_i(\boldsymbol{x})| \geq 2$, we can always find two distinct vertices $j, l \in \mathcal{M}(\boldsymbol{x})$ such that $\lambda x_j + s_j \geq \lambda x_l + s_l$. Let $\hat{\delta} = \min\{x_l, 1 - x_j\}$ and $\boldsymbol{d} = \boldsymbol{e}_j - \boldsymbol{e}_l$. Since $\boldsymbol{x} \in [0,1]^n$ and $|\mathcal{M}_i(\boldsymbol{x})| \leq 1$, $\forall i \in [r]$, we have $\sum_{l \in \mathcal{C}_i \setminus \mathcal{M}_i(\boldsymbol{x})} x_l \geq k_i$, $\forall i \in [r]$, which implies that $\boldsymbol{x} + \delta\boldsymbol{d}$, for every $\delta \in (0, \hat{\delta}]$, is still feasible of (7). Similar to (14), we can obtain $g(\boldsymbol{x} + \delta\boldsymbol{d}) - g(\boldsymbol{x}) > 0$, for every $\delta \in (0, \hat{\delta}]$, which implies that $\boldsymbol{d}$ is an ascent direction at $\boldsymbol{x}$.

Therefore, there always exists an ascent direction at $\boldsymbol{x}$, which implies that $\boldsymbol{x}$ is not a local maximizer of (7). □

## D  Proof of Theorem 4

*Proof.* The proof follows the same argument as Theorem 5 in Lu et al. (2025a), with only minor differences in notation and the extension from the feasible set of D$k$S to that of VAC-D$k$S. As the underlying structure is preserved, the original proof applies directly. We include the adapted version here for completeness.

Since $\boldsymbol{x}$ is a local maximizer of (7) with the diagonal loading parameter $\lambda_1$, there exists $\epsilon > 0$ such that

$$
\boldsymbol{x}^\mathsf{T}(\boldsymbol{A} + \lambda_1\boldsymbol{I})\boldsymbol{x} \geq \boldsymbol{y}^\mathsf{T}(\boldsymbol{A} + \lambda_1\boldsymbol{I})\boldsymbol{y},
\tag{15}
$$

for every $\boldsymbol{y} \in \mathcal{D}_\epsilon$, where $\mathcal{D}_\epsilon = \{\boldsymbol{y} \in \mathcal{D}_k^n \cap \mathcal{F} \mid \|\boldsymbol{x} - \boldsymbol{y}\|_2 \leq \epsilon\}$.

We aim to show that the same inequality holds when the diagonal loading parameter is increased to $\lambda_2 > \lambda_1$. From Lemma 1, we know that $\boldsymbol{x}$ is integral. Then for any $\boldsymbol{y} \in \mathcal{D}_\epsilon$, we have

$$
\begin{aligned}
&\boldsymbol{x}^\mathsf{T}(\boldsymbol{A} + \lambda_2\boldsymbol{I})\boldsymbol{x} - \boldsymbol{y}^\mathsf{T}(\boldsymbol{A} + \lambda_2\boldsymbol{I})\boldsymbol{y} \\
=&\boldsymbol{x}^\mathsf{T}(\boldsymbol{A} + \lambda_1\boldsymbol{I})\boldsymbol{x} + (\lambda_2 - \lambda_1)\|\boldsymbol{x}\|_2^2 - \boldsymbol{y}^\mathsf{T}(\boldsymbol{A} + \lambda_1\boldsymbol{I})\boldsymbol{y} - (\lambda_2 - \lambda_1)\|\boldsymbol{y}\|_2^2 \\
\geq&(\lambda_2 - \lambda_1)(\|\boldsymbol{x}\|_2^2 - \|\boldsymbol{y}\|_2^2) \\
\geq&0,
\end{aligned}
\tag{16}
$$

where the first inequality follows from the local optimality of $\boldsymbol{x}$ under $\lambda_1$ and the last inequality holds because $\|\boldsymbol{z}\|_2^2$ is maximized over $\mathcal{D}_k^n \cap \mathcal{F}$ when $\boldsymbol{z}$ is integral.

Therefore, $\boldsymbol{x}$ remains a local maximizer of (7) with the diagonal loading parameter $\lambda_2$. □

## E  Proof of Theorem 5

*Proof.* The first term in (11) is due to the fact that there are at most $\frac{k(k-1)}{2}$ edges in the graph and the edge weight is at most $w_{\max}$.

The second term in (11) can be derived from

$$
\begin{aligned}
\frac{\boldsymbol{x}_Q^{*\mathsf{T}}\boldsymbol{A}\boldsymbol{x}_Q^*}{w_{\max}k(k-1)} &\leq \frac{\boldsymbol{x}_B^{*\mathsf{T}}\boldsymbol{A}\boldsymbol{y}_B^*}{w_{\max}k(k-1)} = \frac{\boldsymbol{x}_B^{*\mathsf{T}}\boldsymbol{A}_1\boldsymbol{y}_B^*}{w_{\max}k(k-1)} + \frac{\boldsymbol{x}_B^{*\mathsf{T}}(\boldsymbol{A} - \boldsymbol{A}_1)\boldsymbol{y}_B^*}{w_{\max}k(k-1)} \\
&\leq \frac{\boldsymbol{x}^{*\mathsf{T}}\boldsymbol{A}_1\boldsymbol{y}^*}{w_{\max}k(k-1)} + \frac{\boldsymbol{x}_B^{*\mathsf{T}}(\boldsymbol{A} - \boldsymbol{A}_1)\boldsymbol{y}_B^*}{w_{\max}k(k-1)} \leq \frac{\boldsymbol{x}^{*\mathsf{T}}\boldsymbol{A}_1\boldsymbol{y}^*}{w_{\max}k(k-1)} + \frac{\sigma_2(\boldsymbol{A})}{w_{\max}(k-1)},
\end{aligned}
\tag{17}
$$

where $\boldsymbol{x}_Q^*$ is an optimal solution to the quadratic optimization problem (1) and $(\boldsymbol{x}_B^*, \boldsymbol{y}_B^*)$ is an optimal solution to the following bilinear optimization problem

$$\max_{\boldsymbol{x}, \boldsymbol{y} \in \mathcal{B}_k^n \cap \mathcal{F}} \boldsymbol{x}^\top \boldsymbol{A} \boldsymbol{y}. \tag{18}$$

The third term in (11) can be derived from

$$\frac{\boldsymbol{x}_Q^{*\top} \boldsymbol{A} \boldsymbol{x}_Q^*}{w_{\max} k(k-1)} \leq \frac{\sigma_1(\boldsymbol{A})}{w_{\max}(k-1)}, \tag{19}$$

where $\boldsymbol{x}_Q^*$ is an optimal solution to the quadratic optimization problem (1). □

## F   Additional Experimental Results

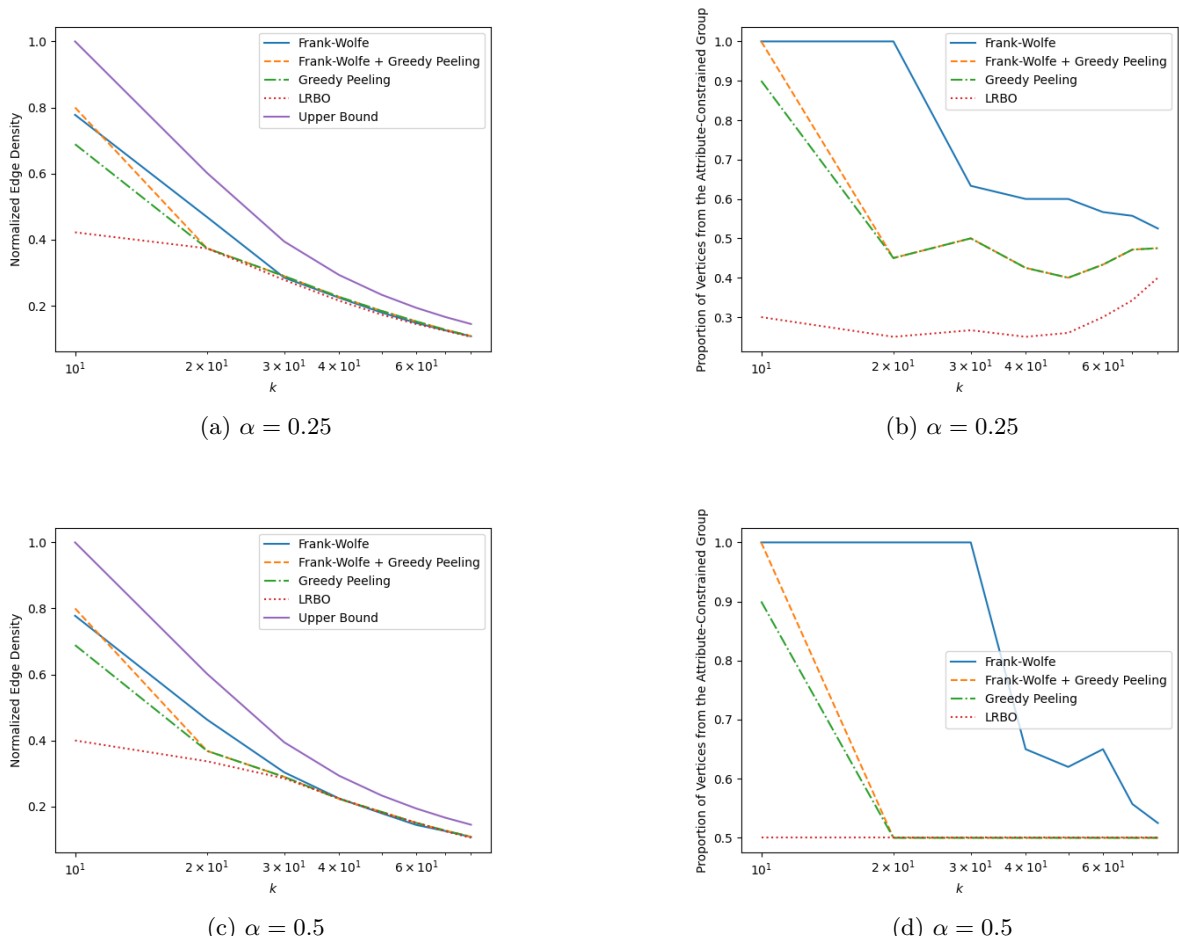

(a) $\alpha = 0.25$

(b) $\alpha = 0.25$

(c) $\alpha = 0.5$

(d) $\alpha = 0.5$

Figure 3: Normalized edge density and attribute-constrained group proportion on the Books dataset under different $\alpha$. Result: Frank–Wolfe variants (blue and orange) outperform other methods in terms of density for small $k$ and all methods have similar performance for large $k$.

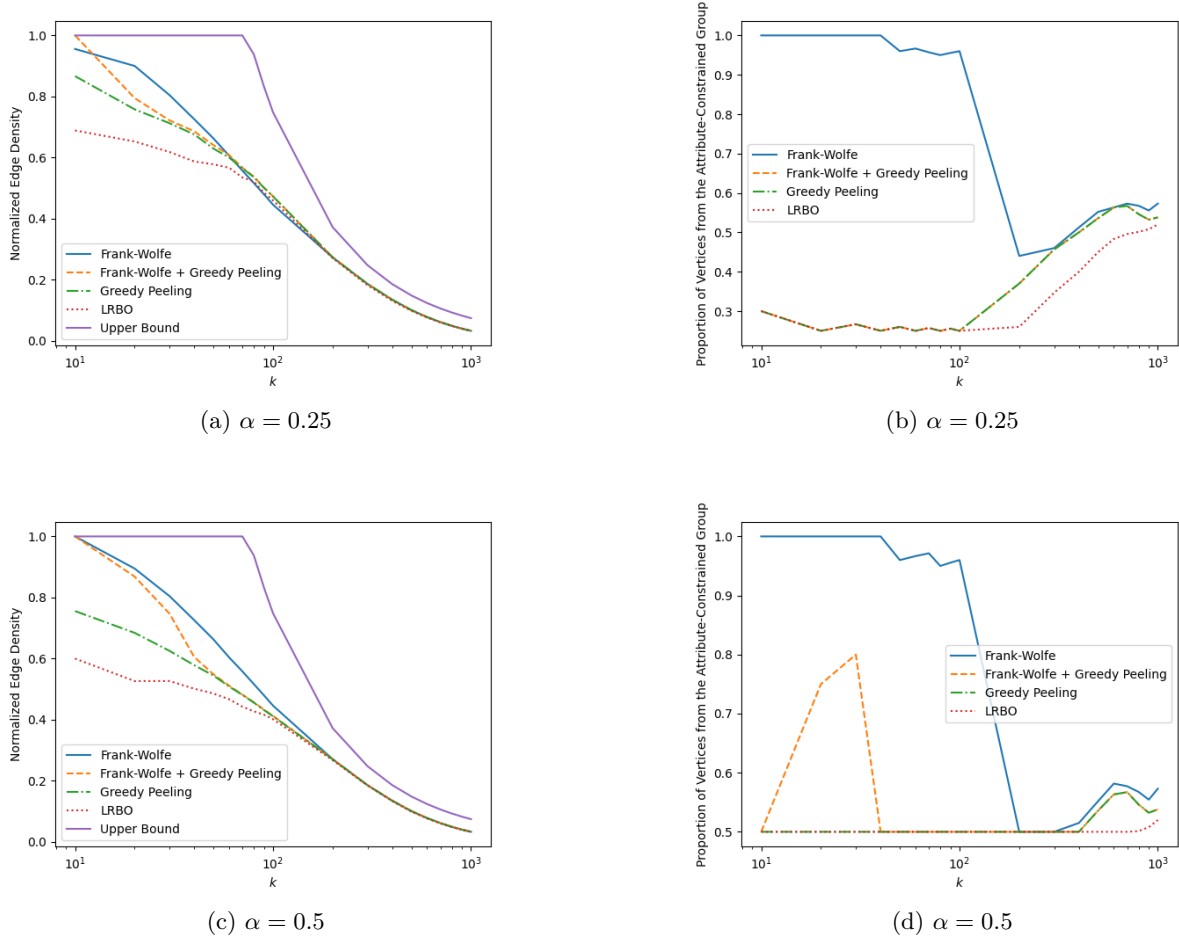

Figure 4: Normalized edge density and attribute-constrained group proportion on the Blogs dataset under different $\alpha$. Result: Frank–Wolfe variants (blue and orange) outperform other methods in terms of density for small $k$ and all methods have similar performance for large $k$.

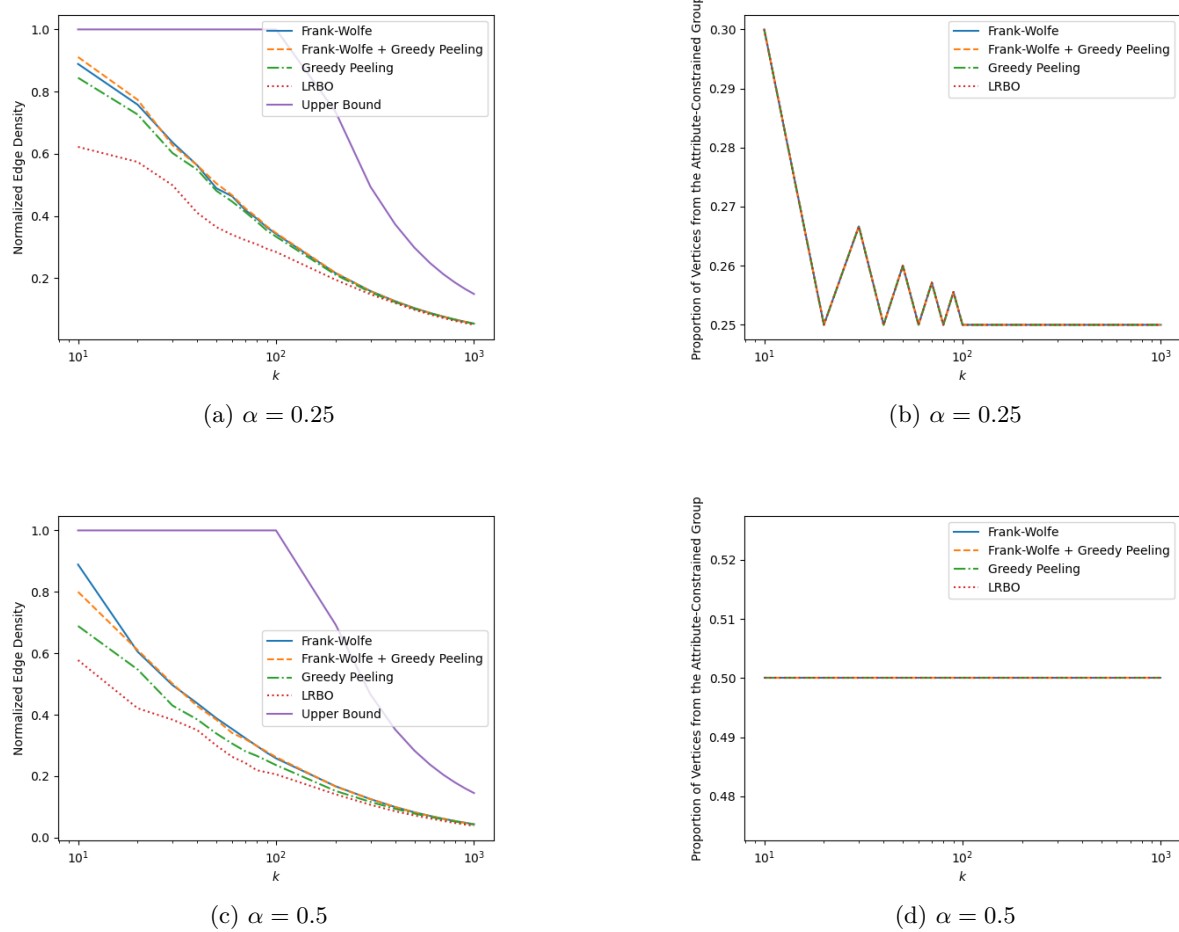

Figure 5: Normalized edge density and attribute-constrained group proportion on the GitHub dataset under different $\alpha$. Result: Frank–Wolfe variants (blue and orange) outperform other methods in terms of density for small $k$ and all methods have similar performance for large $k$.

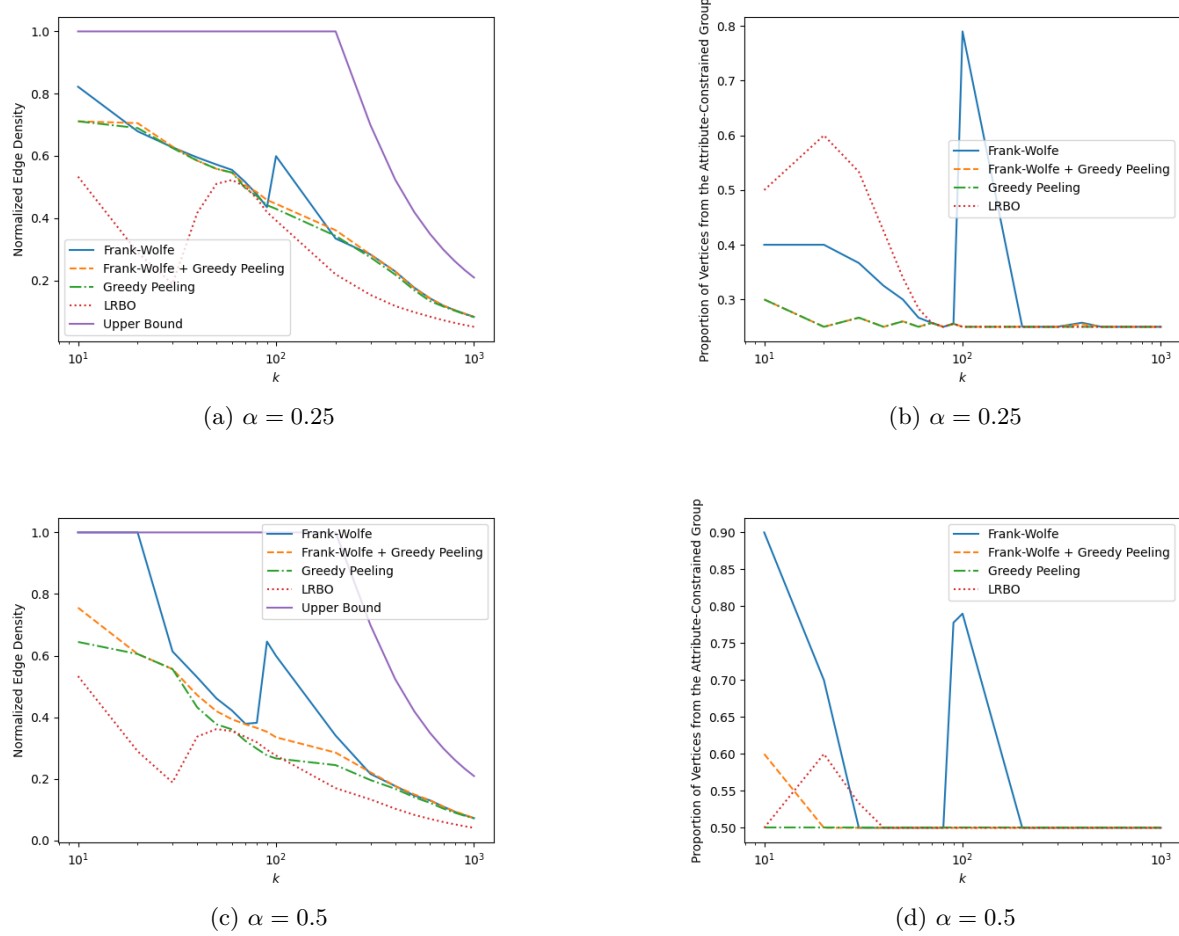

Figure 6: Normalized edge density and attribute-constrained group proportion on the Wikipedia dataset under different $\alpha$. Result: Frank–Wolfe variants (blue and orange) outperform other methods in terms of density across most of $k$.

Table 6: Greek MPs and media outlets identified by LFPR$_N$ from the Greek Politics dataset. Labels indicate political leanings (**(0)**: left leaning, **(1)**: right leaning).

| Rank | Name |
|:---:|:---:|
| 1. | Andreas Loverdos (0) |
| 2. | *Ta Nea* (0) |
| 3. | Kostas Skandalidis (0) |
| 4. | Fofi Gennimata (0) |
| 5. | Evi Christofilopoulou (0) |
| 6. | Giannis Maniatis (0) |
| 7. | Odysseas Konstantinopoulos (0) |
| 8. | *LiFO* (0) |
| 9. | Evangelos Venizelos (0) |
| 10. | Notis Mitarachi (1) |
| 11. | Theodoros Karaoglou (1) |
| 12. | *News 24/7* (0) |
| 13. | Nikitas Kaklamanis (1) |
| 14. | *Efimerida Empros* (0) |
| 15. | Giorgos Koumoutsakos (1) |
| 16. | Dora Bakoyannis (1) |
| 17. | Spyros Lykoudis (0) |
| 18. | *Real News* (1) |
| 19. | *Proto Thema* (1) |
| 20. | Kostis Hatzidakis (1) |

