# OpenReview forum: "The Vertex-Attribute-Constrained Densest $k$-Subgraph Problem"
_TMLR — Accepted by TMLR_

### Review · Reviewer_uq3V · 2025-08-30

**Summary Of Contributions:**

**Summary**

This paper suggests a new variant of the densest k-subgraph (DkS) problem. This new variant adds a Vertex-Attribute-Constraint (VAC) to DkS. Compared with the vanilla DkS, the VAC version additionally gives as input a parameter r and k_1, ..., k_r, each data point is given a group ID in [r], and the constraint asks to find a subset that has *exactly* k vertices and also contains at least k_i vertices from the i-th group. The goal is still to maximize the density.

This problem is clearly harder than the vanilla DkS which is already hard. This paper suggests a new mathematical programming formulation for the problem, and it is shown that this is a relaxation and that it is tight, in the sense that every optimal solution for the original problem is also optimal for the relaxation. Based on this formulation, heuristic algorithms are proposed to solve VAC-DkS. Experiments are conducted to validate the performance of the new algorithms.

**Strength**
- I find the new formulation intuitive, and the new heuristics makes sense to me
- The experiments are comprehensive

**Weakness**
- The suggested algorithms do not seem to have worst-case guarantee
- The techniques are based on unconstrained version, and no new framework is proposed specifically for the constraint, which may suggest that the approaches are suboptimal

**Audience:**

Yes

**Audience Explanation:**

The DkS problem is a fundamental problem, and similar variants have been considered. This new variant is certainly interesting to people who cares about DkS and its applications.

**Broader Impact Concerns:**

None.

**Claims And Evidence:**

No

**Claims Explanation:**

I find several important aspects not clearly discussed. I would list the most severe issues here, and other issues are listed under "requested changes".

- You defined the tightness as "a solution that is optimal for the original problem is also an optimal solution to the relaxed problem". Does this imply that there is no integrality gap, i.e., the optimal solution for the relaxed problem equals to that of the original? I find this too strong, because it is even NP-hard to find the optimal value.

- You mentioned a paper: "Qiheng Lu, Nicholas D. Sidiropoulos, and Aritra Konar. On densest k-subgraph mining and diagonal loading. arXiv preprint arXiv:2410.07388, 2024." I find the content of the paper very similar to this submission. How are they related? Also, what concerns me is that you seem to suggest that the notion of tightness comes from this said paper, so it's important to understand the relationship.

- It is also unclear how your new variant compares with existing similar ones. The experiment result also requires a more detailed interpretation.

**Requested Changes:**

Besides the major issues listed above, addressing the following questions are also important for my recommendation for acceptance/rejection.

- The related work section mentions several works on VAC DkS. How does your setting compare with them? I don't find a conclusive discussion.

- In the experiments, I don't find conclusions for each individual experiment. Only a factual explanation for each baseline/algorithm is given. This makes it difficult to know how well your algorithm performs. In fact, it is even unclear which curve is your new algorithm and which is the baseline.

- In the experiments, the modeling/parameters of the synthesized datasets are not well explained, and they seem to be quite "magic" to me.

- Section 4.2, you mentioned you derived an upper bound. How is this done?

---

> ### Author Response · Authors · 2025-09-01
> **Response to Reviewer uq3V (Part I)**
>
> We would like to thank the reviewer for the review comments. Please find our point-by-point responses below.
>
> >**Comment 1**: The suggested algorithms do not seem to have worst-case guarantee.
>
> **Response**: Indeed, our proposed algorithm does not have a worst-case approximation guarantee. As shown in Theorem 1, VAC-D$k$S is at least as difficult to approximate as the classical D$k$S. The best-known polynomial-time approximation algorithm for D$k$S only achieves a ratio of $O(n^{1/4 + \epsilon})$, which is often too loose to provide a meaningful guarantee of practical solution quality. Given these significant theoretical barriers to achieving a useful approximation ratio, our paper pursues an alternative approach. Instead of focusing on the algorithm's worst-case performance, we provide a rigorous theoretical analysis of the problem's formulation itself. We first prove that a continuous relaxation of VAC-D$k$S is provably tight, i.e., the optimal solutions to the original problem remain optimal for the relaxed problem. We also provide an insightful analysis of the optimization landscape of relaxed problem, which reveals that increasing the diagonal loading parameter beyond the minimum required for tightness makes this landscape more challenging. Therefore, we justify our approach not with a traditional approximation ratio, but by proving that our formulation is tight and its optimization landscape is well-behaved, making it highly amenable to efficient algorithms.
>
> >**Comment 2**: The techniques are based on unconstrained version, and no new framework is proposed specifically for the constraint, which may suggest that the approaches are suboptimal.
>
> **Response**: We note that our formulation is an extension of classical D$k$S, which is a constrained problem to begin with. While we build upon the powerful continuous relaxation framework for D$k$S, the new vertex-attribute constraints introduce significant theoretical challenges. Our key theoretical novelty is a more sophisticated two-stage rounding technique developed specifically to handle these new constraints, which is essential to prove the tightness and analyze the landscape of our formulation. Such a framework is designed specifically to deal with the vertex-attribute constraints, and it is not a straightforward extension of the continuous relaxation framework for D$k$S.
>
> Furthermore, Frank–Wolfe is a suitable choice for this problem. This is because the exact solution of the linear maximization subproblem remains simple despite the newly introduced attribute constraints. This makes the proposed Frank–Wolfe approach natural and elegant in its simplicity.
>
> >**Comment 3**: You defined the tightness as "a solution that is optimal for the original problem is also an optimal solution to the relaxed problem". Does this imply that there is no integrality gap, i.e., the optimal solution for the relaxed problem equals to that of the original? I find this too strong, because it is even NP-hard to find the optimal value.
>
> **Response**: The reviewer is correct that our definition of tightness implies no integrality gap. However, we respectfully clarify that this claim is not "too strong" and does not contradict the problem's NP-hardness. The key point is that our relaxed continuous formulation—maximizing a non-concave function—is itself still NP-hard to solve to global optimality. This is why our landscape analysis is crucial. The relatively benign optimization landscape of our formulation makes it easier for gradient-based algorithms, such as Frank–Wolfe, to find high-quality solutions.
>
> >**Comment 4**: You mentioned a paper: "Qiheng Lu, Nicholas D. Sidiropoulos, and Aritra Konar. On densest k-subgraph mining and diagonal loading. arXiv preprint arXiv:2410.07388, 2024." I find the content of the paper very similar to this submission. How are they related? Also, what concerns me is that you seem to suggest that the notion of tightness comes from this said paper, so it's important to understand the relationship.
>
> **Response**: The paper by Lu et al. introduced a framework for unweighted classical D$k$S. This submission represents a significant generalization by introducing, motivating, and solving the more general weighted VAC-D$k$S. The new attribute constraints in this more general setting render the proof techniques from Lu et al. insufficient, requiring a key theoretical novelty of this work: a more sophisticated two-stage rounding technique to prove the tightness and analyze the optimization landscape.

---

> ### Author Response · Authors · 2025-09-01
> **Response to Reviewer uq3V (Part II)**
>
> >**Comment 5**: It is also unclear how your new variant compares with existing similar ones. The experiment result also requires a more detailed interpretation.
>
> **Response**: Regarding the comparison to existing variants, in Section 1.1, we survey prior works and in Section 2, directly following our problem definition, we provide a detailed, point-by-point comparison with existing formulations. The fundamental distinction, as we discuss there, is the novel D$k$S-based formulation versus prior DSG-based methods. This foundational difference, which provides advantages like explicit size control, makes the two types of approaches not directly comparable. Furthermore, compared with prior formulations, our approach provides greater flexibility in controlling group composition while ensuring meaningful lower bounds on group proportions.
>
> Regarding a more detailed interpretation of the experimental results, for instance, our algorithm's success in recovering the planted clique in the high-noise setting (Table 2), where Greedy Peeling and LRBO completely fail, offers strong empirical validation of our theoretical analysis. It demonstrates that a well-structured continuous formulation with a benign landscape, such as ours, can be significantly more effective. we have included additional discussion in Section 5.3 in the revised manuscript.
>
> >**Comment 6**: The related work section mentions several works on VAC DkS. How does your setting compare with them? I don't find a conclusive discussion.
>
> **Response**: We begin with a detailed survey of prior work and its limitations in Section 1.1. Following that, we summarize the key advantages of our formulation over existing approaches in our list of contributions in Section 1.2. Finally, directly after the formal problem definition in Section 2, we present a detailed, point-by-point comparison. The takeaway from these discussions is the fundamental difference between our D$k$S-based formulation and prior DSG-based formulations. We place our most detailed comparison in Section 2 because a truly conclusive comparison can only be made after our new formulation has been precisely defined.
>
> >**Comment 7**: In the experiments, I don't find conclusions for each individual experiment. Only a factual explanation for each baseline/algorithm is given. This makes it difficult to know how well your algorithm performs. In fact, it is even unclear which curve is your new algorithm and which is the baseline.
>
> **Response**: Our curve in Figures 1–2 in the main paper and Figures 3–6 in Appendix F is the blue one (labeled as Frank–Wolfe) in the legend. It is clear that it outputs the highest quality solutions. We have summarized our findings in the caption of these figures in the revised manuscript to make it easier to find the take-home points from each experiment.
>
> >**Comment 8**: In the experiments, the modeling/parameters of the synthesized datasets are not well explained, and they seem to be quite "magic" to me.
>
> **Response**: The parameters for our synthetic datasets were deliberately chosen to test distinct challenges, as described in our paper. The first setting (Table 2) was designed to test algorithmic robustness against significant background noise. In contrast, the second setting (Tables 3 and 4) was chosen to specifically evaluate the scalability and runtime efficiency of the algorithms on weighted and unweighted large-scale graphs.
>
> >**Comment 9**: Section 4.2, you mentioned you derived an upper bound. How is this done?
>
> **Response**: The upper bound is a generalization of an upper bound for D$k$S proposed by Papailiopoulos et al. and the proof of this upper bound is provided in Appendix E.
>
> Thank you again for your review.

---

> > ### Comment · Reviewer_uq3V · 2025-11-09
> >
> > Thanks for the detailed response.
> >
> > * For the "tightness" of relaxation, it makes more sense if the relaxation is still NP-hard. But your wording in the paper is not clear. Reading your sentence in the second bullet of "1.2 Our Contributions": "**Although** the VAC-DkS problem is NP-hard, we prove that a natural relaxation is tight and analyze the optimization landscape of the relaxed problem" -- you said "although it is NP-hard", does this NP-hardness have anything to do with whether the relaxation can be tight? I thought you want to say "although it is NP-hard, we can have some relaxation that is not NP-hard"?
> >
> > * Actually, why do we need that the relaxation is tight? I don't find this result used anywhere in the paper. Is it important on its own, since it is NP-hard to solve anyway, and hence no approximation algorithm with ratio guarantee may be based on it?
> >
> > * You mentioned your technical contribution is a sophisticated rounding process (in the second bullet of Sec 1.2). Which part in the paper is this from, is this Theorem 3? Is a similar process considered in previous works, and would you do a technical comparison?
> >
> > * I observed in the experiments results, the Frank-Wolfe curve is very much non-smooth and large fluctuations are observed. Is there an explanation of this?
> >
> > * I'm still not sure about the relation to the previous papers. Specifically, I'm talking about two: one is the arxiv paper I listed, and another AAAI 2025 paper which you also cited, and they are very similar to the current submission. I think your current submission builds heavily upon both papers, albeit it solves a constrained version of the problem. The heavy-lifting of the techniques seems to come from both previous papers. I guess a question here is, whether the two previous papers are considered"preliminary version" of this journal submission, or this journal submission is completely independent to the two papers. From the current presentation, it seems to be the later case, but I suspect it is the former case. It's good to clarify, and this helps to justify the contribution. Also, if you claim it is independent, then I would suggest to do a more thorough comparison -- for example, would you adapt their algorithm as a baseline in the experiments?

---

> > > ### Author Response · Authors · 2025-11-10
> > > **Response to Follow-up Questions from Reviewer uq3V (Part I)**
> > >
> > > We would like to thank the reviewer for the follow-up questions. Please find our point-by-point response below.
> > > >**Question 1**: For the "tightness" of relaxation, it makes more sense if the relaxation is still NP-hard. But your wording in the paper is not clear. Reading your sentence in the second bullet of "1.2 Our Contributions": "Although the VAC-DkS problem is NP-hard, we prove that a natural relaxation is tight and analyze the optimization landscape of the relaxed problem" -- you said "although it is NP-hard", does this NP-hardness have anything to do with whether the relaxation can be tight? I thought you want to say "although it is NP-hard, we can have some relaxation that is not NP-hard"?
> > >
> > > **Response**: What we meant to convey was that we start from an NP-hard problem, and then when we apply an "intuitive" continuous relaxation on the problem, it surprisingly turns out that this is tight. We wanted to highlight the non-obvious nature of this fact. However, we did not word this as precisely as we would have liked, and hence, one can also reach the conclusion the reviewer has made. To eliminate this ambiguity, we will revise the sentence to be a more direct statement of fact. The proposed revision: We prove that the VAC-D$k$S problem is NP-hard. We also prove that a natural continuous relaxation is tight and analyze the optimization landscape of the relaxed problem.
> > >
> > > >**Question 2**: Actually, why do we need that the relaxation is tight? I don't find this result used anywhere in the paper. Is it important on its own, since it is NP-hard to solve anyway, and hence no approximation algorithm with ratio guarantee may be based on it?
> > >
> > > **Response**: The suboptimality in using a continuous optimization algorithm to solve a NP-hard combinatorial problem (like VAC-D$k$S) comes from two sources:
> > > 1. **Relaxation Gap**: The gap introduced because the original problem's optimal solution is not optimal for the relaxed problem (e.g., as in many convex relaxations).
> > > 2. **Optimization Error**: The error introduced because the optimization algorithm does not find the global optimum.
> > >
> > > The importance of our theoretical results in Section 3 is that they provide a unified way to minimize both sources of suboptimality simultaneously: the tightness (Corollary 1) eliminates the relaxation gap and the landscape analysis (Lemma 1 and Theorem 4) reduces the optimization error.
> > >
> > > In summary, tightness is crucial because it ensures our relaxed formulation is sound, and our landscape analysis shows this formulation is also amenable to gradient-based algorithms.
> > >
> > > >**Question 3**: You mentioned your technical contribution is a sophisticated rounding process (in the second bullet of Sec 1.2). Which part in the paper is this from, is this Theorem 3? Is a similar process considered in previous works, and would you do a technical comparison?
> > >
> > > **Response**: The reviewer is correct that the proof of Theorem 3 is a primary example of this more sophisticated rounding technique. This technique is actually the fundamental proof strategy that we used in the proofs for Theorem 2 (Appendix A), Theorem 3 (Appendix B), and Lemma 1 (Appendix C). We had to develop this strategy to handle the main challenge of our paper, which is the introduction of new attribute constraints.
> > >
> > > The "sophistication" lies in the contrast with the proof for the unconstrained D$k$S (Lu et al., 2024). Their proof only needs to preserve one constraint ($\sum_{i \in [n]} x_{i} = k$), while our proof must simultaneously preserve $r + 1$ constraints ($\sum_{i \in [n]} x_{i} = k$ and $r$ attribute constraints). To achieve this, we developed a two-stage proof strategy:
> > > 1. **Intra-Group Rounding**: First, for any non-integral point $\boldsymbol{x}$, we round its non-integral entries within a single attribute group. We repeat this process until we arrive at a (still potentially non-integral) $\hat{\boldsymbol{x}}$ where at most one non-integral entry exists per attribute group.
> > > 2. **Inter-Group Rounding (The Key Insight)**: After the first stage, we might still have non-integral entries, but they are now in different groups (e.g., non-integral entries $\hat{x}\_{i}$ and $\hat{x}\_{j}$ are in group 1 and group 2, respectively). Because $\hat{x}\_{i}$ is the only non-integral entry left in group 1, we know that the other integral entries in group 1 must already be satisfying group 1's attribute constraint ($\sum_{l \in C_{1} \setminus \\{i\\}} x_{l} \geq k_{1}$). This means that the non-integral entry $\hat{x}\_{i}$ is free from group 1's attribute constraint. The same logic applies to $\hat{x}\_{j}$ in group 2. Therefore, we can safely round the non-integral entries in different groups.
> > >
> > > This two-stage strategy constitutes the non-trivial theoretical generalization that is required for VAC-D$k$S.

---

> > > ### Author Response · Authors · 2025-11-10
> > > **Response to Follow-up Questions from Reviewer uq3V (Part II)**
> > >
> > > >**Question 4**: I observed in the experiments results, the Frank-Wolfe curve is very much non-smooth and large fluctuations are observed. Is there an explanation of this?
> > >
> > > **Response**: We thank the reviewer for this observation. The fluctuation is an expected consequence and stems from two distinct sources:
> > > 1. **The Inherent Nature of the Problem**: First, the ground-truth optimal density can exhibit sharp drops when a change in $k$ forces the optimal solution to incorporate a new, less-dense set of nodes.
> > > 2. **The Non-Convex Formulation**: Second, we are solving a non-convex relaxed problem. As the parameter $k$ changes, the entire optimization landscape and the initialization both change. As a result, our algorithm converges to different solutions.
> > >
> > > >**Question 5**: I'm still not sure about the relation to the previous papers. Specifically, I'm talking about two: one is the arxiv paper I listed, and another AAAI 2025 paper which you also cited, and they are very similar to the current submission. I think your current submission builds heavily upon both papers, albeit it solves a constrained version of the problem. The heavy-lifting of the techniques seems to come from both previous papers. I guess a question here is, whether the two previous papers are considered"preliminary version" of this journal submission, or this journal submission is completely independent to the two papers. From the current presentation, it seems to be the later case, but I suspect it is the former case. It's good to clarify, and this helps to justify the contribution. Also, if you claim it is independent, then I would suggest to do a more thorough comparison -- for example, would you adapt their algorithm as a baseline in the experiments?
> > >
> > > **Response**: This submission is a fully independent paper whose contribution is twofold:
> > > 1. We propose a new, valuable problem VAC-D$k$S. As we argue in the introduction and demonstrate in the case study (Section 5.6), this formulation fills a critical gap:
> > > - The classic D$k$S problem ignores attributes, leading to "mundane" results on the Greek politics dataset, whereas VAC-D$k$S finds an interesting subset of politicians.
> > > - Existing attribute-aware formulations are based on the DSG framework, which lacks explicit size control and tends to produce large but loosely connected subgraphs.
> > > 2. We provide a non-trivial theoretical generalization to solve it. As discussed in our answer to the third question, a more sophisticated rounding technique is developed specifically in this paper to handle the new introduced attribute constraints.
> > >
> > > In summary, Lu et al. established a new theoretical paradigm for the classic D$k$S problem. Our paper introduces the new VAC-D$k$S problem and develops the new and non-trivial theoretical tool required to generalize that paradigm to solve it. This is a complete and independent contribution.
> > >
> > > Thank you again for your follow-up questions.

---

### Review · Reviewer_UYw3 · 2025-10-02

**Summary Of Contributions:**

The paper presents a new variant of the densest $k$-subgraph problem, which incorporates constraints on the minimum number of included vertices with each attribute value. The paper then proposes a tight relaxation of the optimization problem and develops an efficient Frank–Wolfe algorithm for solving it. The method is tested on several benchmark datasets and shown to produce more meaningful results than the unconstrained version.

**Audience:**

Yes

**Audience Explanation:**

The densest subgraph problem has a long history and finds applications in many graph learning tasks.

**Claims And Evidence:**

Yes

**Claims Explanation:**

The theoretical and numerical results are presented clearly and demonstrate the usefulness of the proposed method.

**Requested Changes:**

I have the following concerns about the proposed method and the presentation of the paper:

1) The only difference with the previous work Lu et al. (2025) is the newly added constraints. Despite this difference, the theoretical and algorithmic developments are analogous to those in that work. The authors should emphasize more on why the added constraints create difficulties in the theory and algorithm and how these difficulties are resolved in the paper.

2) My major concern with the method is that the relaxed problem (7) is still nonconvex. The complexity analysis in Section 4.1 concerns only the initialization step and the per-iteration cost of the Frank–Wolfe algorithm. Is there any convergence guarantee for the proposed algorithm? Is it possible to further obtain a convex relaxation?

3) In machine learning tasks, one is often faced with noisy or incomplete network samples. In this case, the adjacency matrix may contain incorrect or missing edges. How should one deal with these complications?

Minor comments:
1) When introducing the diagonal loading technique in problem (4), some motivation would be helpful. Why does adding the diagonal matrix improve the solution quality and help guarantee tightness of the relaxation? Is it related to ridge regularization?

2) How is “success count” defined in Tables 2–4? What is considered a success?

---

> ### Author Response · Authors · 2025-10-11
> **Response to Reviewer UYw3 (Part I)**
>
> We would like to thank the reviewer for the constructive feedback. Please find our point-by-point responses below.
>
> >**Comment 1**: The only difference with the previous work Lu et al. (2025) is the newly added constraints. Despite this difference, the theoretical and algorithmic developments are analogous to those in that work. The authors should emphasize more on why the added constraints create difficulties in the theory and algorithm and how these difficulties are resolved in the paper.
>
> **Response**: At a high-level, our approach is analogous to the work of Lu et al. (2025). However, the introduction of vertex-attribute constraints creates a significantly more complex feasible set, which rendered the original proof techniques from Lu et al. (2025) insufficient. Lu et al. (2025) use a rounding procedure on the solution of the continuous relaxation of D$k$S, which looks at a pair of non-integral entries and rounds them appropriately to satisfy the sum-to-$k$ constraint of D$k$S. The main challenge in applying such a simple rounding step for the VAC-D$k$S problem is that rounding two non-integral entries could violate the new attribute constraints. To overcome this, our key technical contribution is a more sophisticated, _two-stage_ rounding procedure. We first perform rounding within each attribute group to ensure at most one non-integral  entry remains per group. We then prove that any remaining non-integral  entries across different groups can be safely rounded without violating the constraints. This novel proof strategy is a non-trivial extension required to handle the more complex problem structure.
>
> >**Comment 2**: My major concern with the method is that the relaxed problem (7) is still nonconvex. The complexity analysis in Section 4.1 concerns only the initialization step and the per-iteration cost of the Frank–Wolfe algorithm. Is there any convergence guarantee for the proposed algorithm? Is it possible to further obtain a convex relaxation?
>
> **Response**: The step size used in Algorithm 2 guarantees that the algorithm converges to a stationary point of (7) (Bertsekas, 2016, p. 268). While the step size proposed by Lacoste-Julien (2016) guarantees a convergence rate of $O(1/\sqrt{t})$, it was empirically shown in Lu et al. (2025) that it has a slower convergence rate than the step size used in Algorithm 2 for the D$k$S problem. For VAC-D$k$S, we observe a similar phenomenon, where the step size used in Algorithm 2 converges much faster in practice than the one proposed by Lacoste-Julien (2016). Establishing stronger convergence guarantees is difficult without making prior assumptions on the problem input, given its intrinsic difficulty.
>
> It is true that a convex relaxation for this problem can be considered, for instance, via extensions of Semidefinite Programming (SDP) or Linear Programming (LP) relaxations of D$k$S. However, such methods necessitate the introduction of additional problem variables. Specifically, for a graph with $n$ vertices, SDP relaxations typically introduce a matrix variable of size $n \times n$, resulting in $O(n^2)$ variables. Solving such an SDP using standard interior-point methods has a complexity of at least $O(n^4)$, making it computationally prohibitive for the large-scale graphs we target. Furthermore, given that VAC-DkS is NP-hard, convex relaxations cannot be tight in general unless P=NP. In contrast, our non-convex relaxation is always tight, computationally tractable, and it comes with landscape analysis and some convergence guarantees.
>
> >**Comment 3**: In machine learning tasks, one is often faced with noisy or incomplete network samples. In this case, the adjacency matrix may contain incorrect or missing edges. How should one deal with these complications?
>
> **Response**: We thank the reviewer for bringing up this practical consideration regarding incorrect or missing edges. Empirically speaking, our approach appears to be robust, judging from our experimental result. For instance, Table 2 shows that compared to baseline methods, our algorithm has a significantly higher probability of recovering the ground-truth planted clique from a graph with significant background noise. A more principled noise-aware version of VAC-D$k$S would be interesting, but beyond the scope of this paper.

---

> ### Author Response · Authors · 2025-10-11
> **Response to Reviewer UYw3 (Part II)**
>
> >**Comment 4**: When introducing the diagonal loading technique in problem (4), some motivation would be helpful. Why does adding the diagonal matrix improve the solution quality and help guarantee tightness of the relaxation? Is it related to ridge regularization?
>
> **Response**: The motivation for using this diagonal loading term is twofold. For the discrete problem (6), the diagonal loading reformulation is an equivalent reformulation. This is because for any such feasible solution, $\lambda \Vert \boldsymbol{x} \Vert_{2}^{2} = \lambda k$, making the added term a constant that does not change the set of optimal solutions. For the continuous problem (7), the $\ell^{2}$ term acts as an incentive to promote the sparsity of the solution. This is because the integral solutions are the points within the feasible set that maximize the $\ell^{2}$-norm. We have added a more detailed explanation for the motivation of the diagonal loading technique in Section 3 in the revised manuscript.
>
> The way this technique improves solution quality is twofold. First, it guarantees the tightness of the relaxation when $\lambda \geq w_{\max}$. Without tightness, there may exist no integral optimal solution. Second, we show how the choice of $\lambda$ affects the optimization landscape, demonstrating that a proper selection makes the resulting landscape more benign, and hence easier for our algorithm to find a high-quality solution.
>
> Although the diagonal loading technique and ridge regularization both incorporate an $\ell^{2}$-norm-squared term into the objective function, their roles in their respective optimization problems are fundamentally different. In ridge regression, the objective is minimized, and the $\ell^{2}$ term acts as a penalty to shrink coefficients toward zero, thus preventing overfitting. By contrast, in our formulation, the objective is maximized, and the $\ell^{2}$ term is used to promote sparsity, thereby guaranteeing the tightness of our relaxation and creating a favorable optimization landscape.
>
> >**Comment 5**: How is "success count" defined in Tables 2–4? What is considered a success?
>
> **Response**: In our synthetic experiments (Tables 2–4), we use the planted clique model where the ground-truth dense subgraph is known. A success is counted if the algorithm recovers the planted ground-truth subgraph. We have added the definition of "success count" in Section 5.1 in the revised manuscript.
>
> Thank you again for your review.

---

### Review · Reviewer_KTgj · 2025-10-15

**Summary Of Contributions:**

The paper has three primary contributions.

First, it introduces a new Vertex-Attribute-based densest k-subgraph formulation (that they define vertex-attribute-constrained-DSG), an extension of the Densest k-subgraph problem. They argue that this framework is more useful/effective compared to existing frameworks as they have explicit control over the subgroup size, whereas the existing vertex-attributed problems can lead to large, loose subgraphs.

Then it shows that the new problem is NP-hard and retains the inapproximability results of the original DSG variant.

The rest of the paper focuses on certain "continuous relaxations", providing solutions through projection-free Frank–Wolfe algorithm as well as analyzing the overall optimization landscape.

**Additional Comments:**

The authors claim that in a political network dataset they make observations that would not have been possible simply with DGS formulation.

Here, one could question the overall applicability of the DGS problem to this question. If we already have the group attributes, why do we want to recover "interesting candidates" through a global DGS based analysis? There could be many alternate ways, such as applying PageRank to the graph and selecting most "influential" nodes from each group, or even recent balanced centrality based measures [1,2] and even unsupervised "balanced ranking" algorithms. Does the DGS based mining provide any novel observation that would not be captured by the aforementioned methods?

Furthermore, how do the existing vertex-attribute-based-DGS problems perform for this problem is an important question that should be addressed.

In summary, while the problem formulation and relaxation seem "intuitive", I am unable to verify why such formulations are important in any data-mining problem.

--- ---

[1] Tsioutsiouliklis S, Pitoura E, Tsaparas P, Kleftakis I, Mamoulis N. Fairness-aware pagerank. InProceedings of the Web Conference 2021 2021 Apr 19 (pp. 3815-3826).

[2] Papasotiropoulos G, Skibski O, Skowron P, Wąs T. Proportional selection in networks. arXiv preprint arXiv:2502.03545. 2025 Feb 5.

[3] Mukherjee CS, Zhang J. Balanced Ranking with Relative Centrality: A multi-core periphery perspective. InThe Thirteenth International Conference on Learning Representations.

**Audience:**

Yes

**Audience Explanation:**

This is an active area of research in data-mining in general, indicating there could be some individuals interested in applying the formulation to their problem.

**Claims And Evidence:**

Yes

**Claims Explanation:**

The paper correctly points out that existing vertex-attribute constrained DSG problems do not focus on exact control on the size of the subgraph.

The proof of this formulation being as hard as DGS is quite simple and follows correctly, and the w.r.t. the third step, analysis of continuous relaxations are done in a rigorous manner, even if they are fairly straightforward.

**Requested Changes:**

I think the paper is well written and self-contained.

Is it possible to test the outcome existing vertex-attribute-based DGS problems on the political network data, and compare the performance? Comparing with only DGS seems incomplete evaluation.

---

> ### Author Response · Authors · 2025-10-20
> **Response to Reviewer KTgj (Part I)**
>
> We would like to thank the reviewer for the constructive feedback. Please find our point-by-point responses below.
>
> >**Comment 1**: The proof of this formulation being as hard as DGS is quite simple and follows correctly, and the w.r.t. the third step, analysis of continuous relaxations are done in a rigorous manner, even if they are fairly straightforward.
>
> **Response**: We thank the reviewer for acknowledging that our analysis was conducted in a rigorous manner. Regarding the analysis being deemed fairly straightforward, we respectfully note that the introduction of attribute constraints in the VAC-D$k$S problem renders the theoretical analysis considerably more challenging than that of the classical D$k$S problem. To address this challenge, we proposed a more sophisticated two-stage rounding technique to handle these attribute constraints.
>
> >**Comment 2**: Is it possible to test the outcome existing vertex-attribute-based DGS problems on the political network data, and compare the performance? Comparing with only DGS seems incomplete evaluation.
>
> **Response**: As we described in Section 1.1, VAC-DSG-based formulations cannot control the size of the detected subgraph, and this makes direct comparison with our approach difficult. For example, in response to the reviewer's comment, we applied the DFSG algorithm of Anagnostopoulos et al. (2020) (using Greedy Peeling for its densest subgraph discovery step) to the 186-vertex Greek Politics dataset. The DFSG algorithm returned a subgraph containing 150 vertices—over 80\% of the entire graph—with a normalized edge weight of only 0.112. This stands in contrast to the compact, 20-vertex subgraph with a high normalized edge weight of 0.391 found by our VAC-D$k$S. This empirically confirms the well-documented tendency of DSG-based formulations to output large but loosely connected subgraphs (Tsourakakis et al., 2013). Such a result fails to identify a cohesive core community. On the other hand, our VAC-D$k$S framework is explicitly designed to find a dense group with a specified size. This experiment demonstrates the practical limitations of existing VAC-DSG-based approaches for this dataset.
>
> >**Comment 3**: Here, one could question the overall applicability of the DGS problem to this question. If we already have the group attributes, why do we want to recover "interesting candidates" through a global DGS based analysis? There could be many alternate ways, such as applying PageRank to the graph and selecting most "influential" nodes from each group, or even recent balanced centrality based measures [1,2] and even unsupervised "balanced ranking" algorithms. Does the DGS based mining provide any novel observation that would not be captured by the aforementioned methods?
>
> **Response**: The reviewer is correct that one could seek "interesting candidates" by ranking influential nodes. However, our primary goal with VAC-D$k$S is not to identify a ranked list of the top-$k$ most important individuals, but rather to identify a cohesive and balanced community. In our Greek Politics case study, the central objective of our framework is to find a subgraph that is simultaneously dense and politically balanced. Specifically, because the edge weights in this dataset represent audience similarity, VAC-D$k$S finds the balanced group of politicians whose followers exhibit the highest degree of overlap. Therefore, our approach is suitable for identifying the core political sphere.
>
> We appreciate the reviewer pointing us to the recent literature on balanced centrality measures [1, 2, 3]. Indeed, these can provide additional insight when applied to the Greek Politics dataset, but they are not as suitable as VAC-D$k$S for identifying the core political sphere. We applied the $\text{LFPR}\_{N}$ of [1] with the allocation parameter $\phi = 0.5$ to the Greek Politics dataset (we set a threshold of 0.2 to convert this weighted graph into an unweighted graph because [1, 2, 3] are designed for unweighted graphs). The output of $\text{LFPR}\_{N}$ is shown in Table 6 in Appendix F. The $\text{LFPR}\_{N}$ Top-20 ranking includes 6 media outlets, whereas our VAC-D$k$S identified a cohesive subgraph consisting exclusively of politicians. This difference highlights the fundamental difference between centrality-based methods (identifying influential entities) and our density-based approach (finding cohesive communities). Furthermore, the subset of politicians with the highest ranks identified by $\text{LFPR}\_{N}$ is highly imbalanced: the 7 highest-ranked politicians are all from the PASOK party, unlike our VAC-D$k$S which can guarantee perfect balance when appropriately parameterized ($k_{1} = k_{2} = k/2$) due to its hard constraints.

---

> ### Author Response · Authors · 2025-10-20
> **Response to Reviewer KTgj (Part II)**
>
> >**Comment 4**: In summary, while the problem formulation and relaxation seem "intuitive", I am unable to verify why such formulations are important in any data-mining problem.
>
> **Response**: Beyond the insights from the Greek politics case study, VAC-D$k$S can address an important need in various data mining domains that require finding a cohesive subgroup of a specific size and composition. Potential applications include:
> - **Team Formation**: Identifying the most cohesive project team of exactly $k$ members that includes required expertise from $r$ different skill groups (represented by attributes). VAC-D$k$S combines the need for precise size control, attribute constraints, and maximization of potential collaboration (modeled as density).
> - **Bioinformatics**: Discovering functional protein complexes (often dense subgraphs) of an estimated size $k$ that must contain specific essential protein types (attributes) to perform their biological role.
>
> The Greek politics case study and potential applications underscore the broader importance and relevance of VAC-D$k$S to the data mining community.
>
> Thank you again for your review.

---

### Author Response · Authors · 2025-10-20
**Response to Editor and Reviewers**

We sincerely thank you for your valuable comments and suggestions on our manuscript. We have carefully addressed all the points raised and have uploaded the revised version of the manuscript for your consideration. We greatly appreciate your time and effort in reviewing our work.

---

### Decision · Action_Editor_6TB1 · 2026-01-14

**Recommendation:** Accept with minor revision

**Audience:**

Yes

**Audience Explanation:**

The densest subgraph problem and its proposed variation are of potential practical interest.

**Claims And Evidence:**

No

**Claims Explanation:**

As pointed out by Reviewer uq3V, there are some outstanding concerns regarding the difference with previous work and the exact nature of the points of departure. and regarding the value of the theoretical contribution vs the practical benefits of the new heuristics. The authors should incorporate the discussions with Reviewer uq3V and give more details and more clarity on the points raised by Reviewer uq3V.

---

### Decision · Action_Editor_Sz4b · 2026-04-15

**Recommendation:** Accept as is

**Additional Comments:**

This paper was reviewed by three expert reviewers. The reviewers raised concerns about several limitations of the method (e.g., it does not provide worst-case guarantees, it may not be optimal since it relies on algorithms for the unconstrained version of the problem). They also expressed concerns about the similarity of the work to that of Lu et al. (2025), the existence of data mining applications that would benefit from this formulation, and the experimental evaluation of the proposed method, particularly the absence of baselines in some experiments.

Following the authors' response and the revision of the manuscript, one reviewer recommended acceptance of the paper, while the other two leaned toward rejection. Based on the reviews, the authors' response, and my own reading of the paper, I believe that the paper's claims are supported by convincing evidence. I agree with the reviewers that there are some weaknesses (e.g., the lack of worst-case guarantees), but the paper does not claim otherwise. Regarding the similarity to the work of Lu et al. (2025), I agree with the reviewers that the submission borrows ideas from that work. However, it addresses a different problem. As for the absence of baselines, the authors argue that there are no directly comparable approaches in the literature, and I find this argument reasonable.

Based on the above, I recommend acceptance of this paper. I encourage the authors to address any remaining concerns raised by the reviewers in the final version.

**Audience:**

Yes

**Audience Explanation:**

Given the long history of the densest subgraph problem in data mining and its continued relevance, the findings of this paper are likely to be of interest to at least some individuals in TMLR's audience.

**Claims And Evidence:**

Yes

**Claims Explanation:**

The main claims made in the submission are the following: (i) a new variant of the Densest $k$-Subgraph problem is introduced that incorporates vertex attribute information; (ii) the new variant is NP-hard, and it is shown that a natural relaxation of it is tight, while the optimization landscape of the relaxed problem is analyzed; (iii) the relaxed problem can be efficiently tackled using a projection-free Frank–Wolfe algorithm; and (iv) the method is effective, can scale to large graphs, and can uncover more meaningful subgraphs than approaches that rely on the classical Densest $k$-Subgraph formulation.

In general, the above claims are supported by accurate and convincing evidence:

Claim (i): The paper proposes a formulation that enables explicit control over the subgraph size and also lower bounds on the number of selected vertices from each attribute group. Therefore, this claim is justified.\
Claim (ii): The paper defines the tightness of a relaxation as follows: a relaxation is tight if every optimal solution to the original problem remains optimal for the relaxed problem. It is theoretically shown that the relaxed problem is indeed tight, and the paper also provides results regarding the optimization landscape of the relaxation.\
Claim (iii): The proposed Frank–Wolfe algorithm can efficiently solve the relaxed problem.\
Claim (iv): This claim is supported by the provided empirical results, although they are not very extensive. The effectiveness of the proposed method, compared to Densest $k$-Subgraph formulations on vertex-attributed networks, is demonstrated in the considered political network mining application.